# Physical and mechanical properties of winter first-year ice in the Antarctic marginal ice zone along the Good Hope Line

Sebastian Skatulla[1], Riesna R. Audh[2,4], Andrea Cook[1], Ehlke Hepworth[2,4], Siobhan Johnson[3,4], Doru C. Lupascu[5], Keith MacHutchon[1], Rutger Marquart[1,4], Tommy Mielke[5], Emmanuel Omatuku[1,4], Felix Paul[5], Tokoloho Rampai[3,4], Jörg Schröder[6], Carina Schwarz[6], and Marcello Vichi[2,4]

[1]Department of Civil Engineering, University of Cape Town, Rondebosch, South Africa
[2]Department of Oceanography, University of Cape Town, Rondebosch, South Africa
[3]Department of Chemical Engineering, University of Cape Town, Rondebosch, South Africa
[4]Marine and Antarctic Research Centre for Innovation and Sustainability (MARIS), University of Cape Town, Rondebosch, South Africa
[5]University of Duisburg-Essen, Institute for Materials Science and Center for Nanointegration Duisburg-Essen (CENIDE), Duisburg-Essen, Germany
[6]University of Duisburg-Essen, Institute of Mechanics, Duisburg-Essen, Germany

**Correspondence:** Sebastian Skatulla (sebastian.skatulla@uct.ac.za)

**Abstract.** As part of the 2019 Southern oCean seAsonal Experiment (SCALE) Winter Cruise of the South African icebreaker SA Agulhas II first-year ice was sampled at the advancing outer edge of the Antarctic marginal ice zone along a $150\,\text{km}$-Good Hope Line transect. Ice cores were extracted from four solitary pancake ice floes of $1.83 - 2.95\,\text{m}$ diameter and $0.37 - 0.45\,\text{m}$ thickness as well as a $12 \times 4\,\text{m}^2$ pancake ice floe of $0.31 - 0.76\,\text{m}$ thickness part of a larger consolidated pack ice domain. The ice cores were subsequently analysed for temperature, salinity, texture, anisotropic elastic properties and compressive strength. All ice cores from both, solitary pancake ice floes and consolidated pack ice, exhibited predominantly granular textures. The vertical distributions of salinity, brine volume and mechanical properties were significantly different for the two ice types. High salinity values of $12.6 \pm 4.9\,\text{PSU}$ were found at the topmost layer of the solitary pancake ice floes but not for the consolidated pack ice. The uniaxial compressive strength for pancake ice and consolidated pack ice were determined as $2.3 \pm 0.5\,\text{MPa}$ and $4.1 \pm 0.9\,\text{MPa}$, respectively. The Young's and shear moduli in longitudinal core direction of solitary pancake ice were obtained as $3.7 \pm 2.0\,\text{GPa}$ and $1.3 \pm 0.7\,\text{GPa}$, and for consolidated pack ice as $6.4 \pm 1.6\,\text{GPa}$ and $2.3 \pm 0.6\,\text{GPa}$, respectively. Comparing Young's and shear moduli measured in longitudinal and transverse core directions, a clear directional dependency was found, in particular for the consolidated pack ice.

## 1 Introduction

The seasonal sea ice advance and retreat in the Antarctic Marginal Ice Zone (MIZ) of the Southern Ocean (SO) is heavily influenced by harsh atmospheric and oceanic conditions. Accordingly, the sea ice dynamics in this region is characterised by high temporal and spatial fluctuations and still elude reliable prediction by current climate models (Hobbs et al., 2016). Due to the limited in situ data from the Antarctic region throughout the year, well established sea ice rheology models such

as Hibler III (1979); Wang and Shen (2010) are phenomenological approaches considering effective properties of spatially variable ice concentration and empirically-derived thickness and mechanical properties. In particular in the Antarctic MIZ, however, the actual sea ice properties and the scale dependency of sea ice deformation are linked to the specific ice types (e.g. consolidated pack ice, solitary pancake ice floes and fluid-like grease ice), varying ice concentration, leads with open water etc. (Dansereau et al., 2016; Rampal et al., 2008; Weiss and Dansereau, 2017)). Moreover, the heterogeneous and highly dynamic sea ice composition is affected by a range of mechanical phenomena in combination such as floe collision dynamics, form drag of interstitial grease ice on pancake ice floes and skin drag which can only be accurately resolved by small-scale models (Alberello et al., 2019; Herman et al., 2019; Kohout et al., 2011; Marquart et al., 2021; Rogers et al., 2016). In this sense, large-scale models are not directly informed by the underlying mechanisms and phenomena defining sea ice behaviour, and the model parameters have little physical meaning. Therefore, a generalization of these models to sea ice conditions other than those for which they were fitted to is problematic (Squire, 2018).

Clearly, more investigation of early-stage ice formation in the Antarctic MIZ is necessary to elucidate the seasonal evolution of the ice morphology and to link the physical and mechanical properties from the sub-millimeter to kilometer-scale, as this stage defines the ice properties also later at the consolidated and melting stages. In order to acquire an in-depth understanding of sea ice characteristics and improve forecasting of sea ice trends, short term and long term, remote observational sea ice data and their reanalysis products need to be cross-referenced with in situ observations comprising the actual degree of ice consolidation, thickness, floe size, material composition and texture. In particular, the mechanics of sea ice in terms of deformation, fracture and faulting is linked to a range of spatial and temporal scales (Weiss and Dansereau, 2017).

On a smaller scale less than $10\,\text{km}$, ice strength is distinctly heterogeneous and anisotropic due to the occurrence of leads (Hutchings and Hibler III, 2008). Also, the distribution of floe characteristics such as size, shape, thickness and concentration becomes significant, in particular in the marginal ice zone (Hutchings et al., 2012; Bennetts et al., 2017; Roach et al., 2018; Marquart et al., 2021). Owing to the rough sea states in the winter SO delaying sea ice consolidation and rafting effects, sea ice composition is vertically changing in terms of grain size, crystallographic texture and fabrics as well as porosity (Timco and Weeks, 2010; Dempsey and Langhorne, 2012). On the meter-scale and smaller, the mechanical behaviour of sea ice comprises elastic, inelastic, brittle and viscous material characteristics which are directly related to its grain and pore structure, brine inclusions as well as the generally temperature and strain rate-dependent material response (Mellor, 1986; Schulson et al., 2006). Short-term and seasonal atmospheric temperature fluctuations, precipitation and associated brine drainage processes are continuously altering sea ice composition and properties (Wells et al., 2011; Galley et al., 2015; Thomas et al., 2020). Accordingly, the mechanical ice properties need to be put into context with ice salinity, temperature, density, porosity, type as well as crystal size and orientation (Nakawo and Sinha, 1981; Kovacs, 1996; Timco and Weeks, 2010). As previously mentioned, in situ data of Antarctic sea ice properties are generally scarce, especially during the winter season due to the challenging access to this region. With regards to physical properties of Antarctic winter sea ice, its salinity (Doble et al., 2003; Eicken, 1992; Tison et al., 2020), density (Urabe and Inoue, 1988b) and texture (Jeffries et al., 1997, 2001; Lange and Eicken, 1991; Tison et al., 2017, 2020) have been studied. However, with regards to its mechanical properties including elasticity and strength as well as their directional dependency a complete lack of data is found, except for uniaxial compression strength of Antarctic

land-fast ice (Urabe and Inoue, 1988b). Accurate knowledge of mechanical ice properties is important to parameterize realistic
small-scale sea ice dynamics models with respect to aspects concerning e.g. the influence of pancake ice floe deformation on
the inelastic collision restitution (Herman et al., 2019), ridging (Yiew et al., 2017) and fracture (Weiss, 2013) as well as the
wave-induced flexural break-up of consolidated pack ice (Passerotti et al., 2022). These mechanical phenomena are strongly
linked to sea ice formation and retreat, sea ice drift and wave attenuation in the Antarctic MIZ (Alberello et al., 2020; Eayrs
et al., 2019; Kohout et al., 2014; Rogers et al., 2016; Smith and Thomson, 2019).

This paper will report on the sampling and testing of winter first-year ice at the edge of the Antarctic marginal ice zone along
the Good Hope Line during the 2019 SCALE[1] Winter Cruise. It is the first winter expedition dedicated to MIZ measurements
in the eastern Weddell Sea region after the earlier observations in the late 1980s, early 1990s and 2013 (e.g. Lange et al. (1989),
Eicken et al. (1994), Kivimaa and Kosloff (1994b), Haas et al. (1992), Doble et al. (2001) and Tison et al. (2017)). The focus is
on the physical and mechanical properties of sea ice in the MIZ from the edge into more consolidated pack ice conditions. The
properties determined include sea ice texture, temperature, bulk salinity, bulk density, elastic properties, uniaxial compressive
strength as well as snow thickness and salinity.

The plan of this paper is as follows: Sec. 2 elaborates on sampling and testing methods, Sec. 3 describes and discusses
physical and mechanical data obtained, and a summary of the findings is provided in Sec. 4.

## 2  Materials and Methods

### 2.1  Sample collection

The 2019 Winter SCALE cruise started with the SA Agulhas II departing from Cape Town on the July 18, 2019 sailing directly
south to the Antarctic MIZ. The ship entered the ice on the July 26, 2019 at 1pm and exited on the July 28, 2019 at 8:30pm.
An overview of the MIZ stations is shown in Fig. 1 with the naming convention and corresponding coordinates listed in Tab. 1.
The station plan was designed to resolve the evolution of sea ice features from the open ocean into more consolidated sea ice
conditions. Due to contingency and time optimization, the sequence of stations did not follow the original design. The naming
convention was extended to preserve the original geographic distribution of MIZ1X being at the edge with the open ocean
and MIZ3X in consolidated pack ice. Stations were therefore turned into clusters, where overboard ice coring, pancake ice
collection, grease ice sampling, buoy deployment, environmental measurements and ocean sampling took place. Additional
biogeochemical sampling was performed at MIZ2X which is, however, not part of this work.

A complete set of sea ice observations was done every hour from the bridge commencing on July 26, 2019 at 1pm and
ended on July 28, 2019 at 8:30pm when entering and exiting the ice, respectively. Observations were collected according to the
Antarctic Sea Ice Processes and Climate (ASPeCt) protocol estimating ice concentration, ice type, floe size and thickness, cloud
cover, visibility, weather, and air and sea surface temperature. The initially higher frequency observations where eventually
combined into the hourly frequency required by ASPeCt (Hepworth et al., 2020).

---

[1]Southern oCean seAsonal Experiment http://scale.org.za/

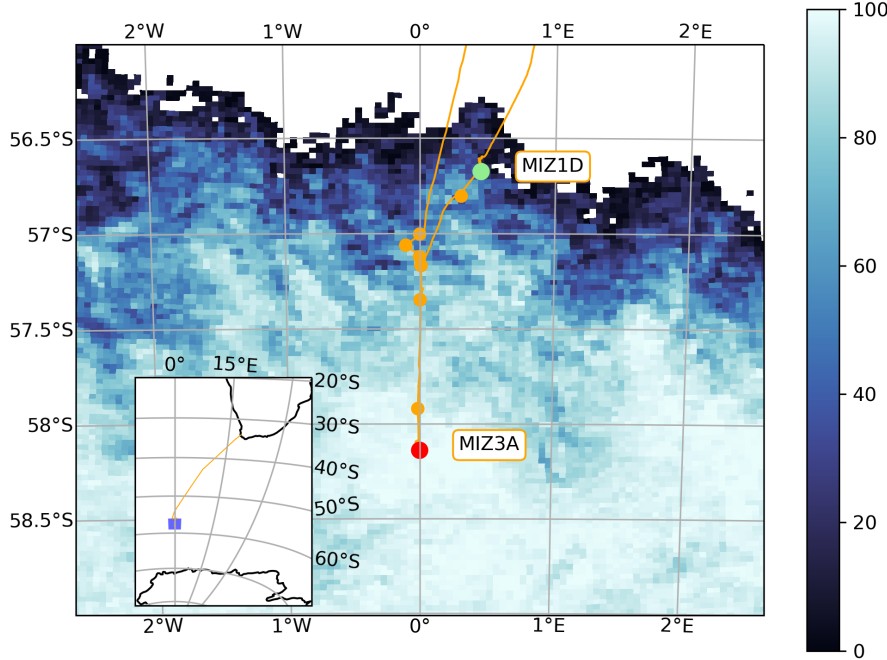

**Figure 1.** Map with the two ice stations MIZ1D and MIZ3A indicated by green and red-coloured circles, respectively. The ship entered the MIZ from the westernmost course and exited from the easternmost course. The sea ice concentration chart was obtained from the University of Hamburg ASI-AMSR2 processing for the July 26, 2019, concentration values in [%]. The small map inlet shows the route from Cape Town to the study area highlighted in blue color.

**Table 1.** List of ice sampling stations.

| Station | Start Date & Time | Latitude | Longitude |
|---------|-------------------|----------|-----------|
| MIZ3A | July 27, 2019 10:38am (UTC) | S 58.13783 | W 0.00442 |
| MIZ1D | July 28, 2019 9:15am (UTC) | S 56.80178 | E 0.30262 |

The first ice floes were observed in the early afternoon of the July 26, 2019. Sea ice features were rapidly varying and large expanses of consolidated pack ice were found at the latitude of MIZ3. Ice navigation and planning was assisted by i) ice edge charting based on satellite data and reanalysis products by the South African Weather Service (SAWS) (de Vos et al., 2021), and ii) a prototype remote sensing product describing multi-year ice concentration and other sea ice types developed by the University of Bremen (Melsheimer and Spreen, 2021). The latter distinguishes the different ice types (young ice, first-year ice and multi-year ice) based on the combination of passive microwave and scatterometer data using an algorithm developed by Environment Canada's Ice Concentration Extractor (ECICE) (Shokr and Agnew, 2013) and corrections by Ye et al. (2015). Fig. 2 shows that the southernmost ice station was expected to be predominantly in first-year ice (FYI) conditions, which was

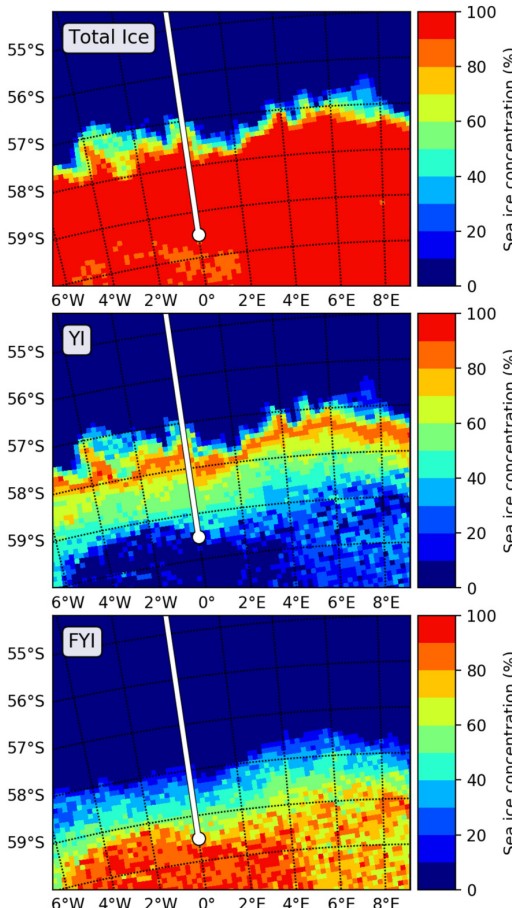

**Figure 2.** Sea ice type concentration data provided by the University of Bremen. Daily concentration of total ice, young ice (YI) and first-year ice (FYI) on the July 24, 2019 prior to the ship entering the MIZ. The cruise plan and the southernmost station (MIZ3A) are also shown. Data are available at https://seaice.uni-bremen.de/data/MultiYearIce

confirmed on site. The MIZ with unconsolidated ice conditions, composed of pancake ice floes of about 1 to 5 m-diameter, likely extended for about 200 km southward as indicated by the young ice (YI) map features in Fig. 2. With reference to WMO (code 3739), ice age ID 5 applied for the southernmost ice station and ice age ID 3 for the most northerly station.

Fieldwork operations involving ice core sampling took place on the southward-bound leg at the most southerly station, MIZ3A, as shown in Fig. 1 on July 27, 2019 between 10am-4pm (UTC). Operations took place at daylight in harsh conditions. The air temperature was $-17\,°C$ and the wind speed $20\,\mathrm{m\,s^{-1}}$, according to the on-board equipment of the South African Weather Services. Large first-year floes, ranging between 500-2000 m in diameter, completely covered the ocean, each one composed of cemented pancake ice floes of smaller sizes. The coring and temperature measurement of cores were done on a single pancake ice floe, about $12\,\mathrm{m} \times 4\,\mathrm{m}$ in dimensions, part of a larger consolidated pack ice domain located off the starboard

bow as shown in Fig. 3. All cores were extracted in vertical direction, perpendicular to the ice floe surface. A total of 26

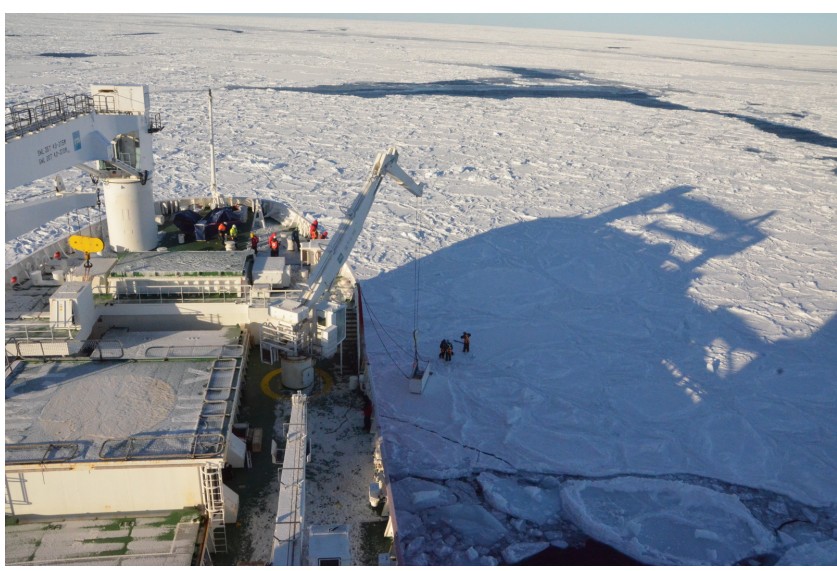

**Figure 3.** Coring field at MIZ3A showing the cemented pancakes with sizes varying between 3 and 15 m.

cores were collected using a 1 m-long 9 cm-diameter Kovacs Mark II barrel operated with an electric drill. Each core was dedicated for a specific type of testing, with the core IDs and corresponding names found in App. A1 in Tab. A1. The length

of the cores is representative of the pack ice thickness. Three of them were immediately processed after coring to determine the vertical temperature distribution. For this purpose, the cores were placed on a plastic rail with markings for drill points 5 cm apart from each other. The temperature probe was put into pre-drilled holes using a cordless drill and the temperature readings were recorded. For temperature measurements, the Testo GMH 3750-GE logger ($\pm 0.03°$ system accuracy) with high precision Pt100 penetration probes GTF 401 1/10 DIN ($\pm 0.03°$ accuracy) were utilized (Testo, Titisee-Neustadt, Germany).

All readings were completed within 4 minutes of coring. After sea ice core collection, a 3D printed Niskin bottle was deployed through two core cavities to retrieve samples of the water at the ice-ocean interface. The Niskin 3D deployment yielded two water samples of approximately 700 ml each from which the boundary layer salinity of the ocean was determined by a 8410A Portasal salinometer ($\pm 0.003$ PSU system accuracy) at room temperature. All 26 cores were put in plastic sleeves, sealed and horizontally stored in crates with a 5 cm-styrofoam insulation for up to 2 hours at environmental temperatures until being

taken aboard for further processing in the glaciological Cold Laboratory or stored in the ship's freezer. The temperature in the laboratory was kept at $-10°$C and the ship's freezer was maintained at $-25°$C, respectively. The cores were continuously kept in a horizontal position during transport and storage to minimize further brine drainage. Keeping the lab temperature on the ship at $-10°$C has been considered a good compromise by the authors so that the large amount of ice samples could be processed as quickly as possible. During processing of ice specimens in the Cold Laboratory, no noticeable brine loss has

occurred.

The collection of solitary pancake ice floes took place on the northward-bound leg at the second most northerly station, MIZ1D, on July 28, 2019 between 10:45am-4pm (UTC). Station MIZ1D was characterized by 80% of the ocean surface covered by pancake ice floes of about 1-3 m diameter as shown in Fig. 4. The relatively warmer air temperature and lower wind speed were recorded at $-8\,°\mathrm{C}$ and $12\,\mathrm{m\,s^{-1}}$, respectively, and a dampened swell was observed. The calm ocean and wind conditions allowed for the quick selection and isolation of pancake ice floes. The pancakes were collected from the ocean using the helideck crane and a custom-made $5 \times 5\,\mathrm{m^2}$ heavy-duty net held by a spreader beam construction designed for a maximum payload of $3.5\,\mathrm{t}$.

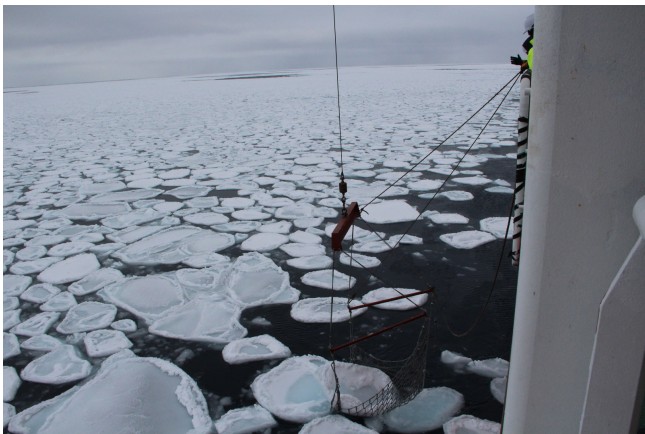

**Figure 4.** Sampling of solitary pancake ice floes with diameters of about $2.5\,\mathrm{m}$ using the ship's crane.

Four pancake ice floes, labelled A to D, were collected and placed onto individual wooden grids as depicted in Fig. 5 to ease coring and prevent the contamination by the ship deck. Details on the different pancake dimensions are given in Tab. 2.

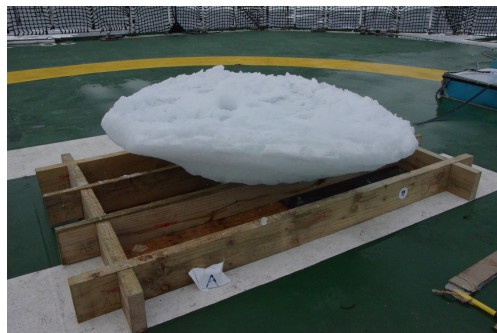
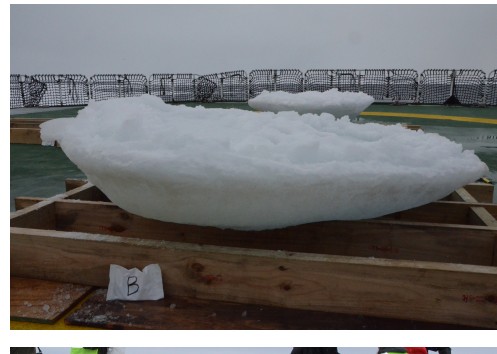
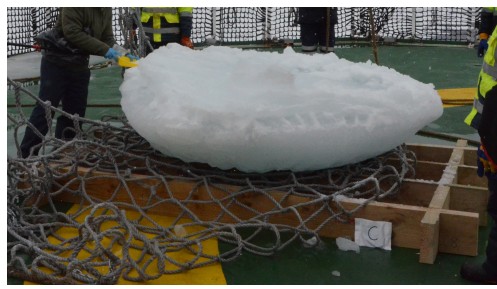
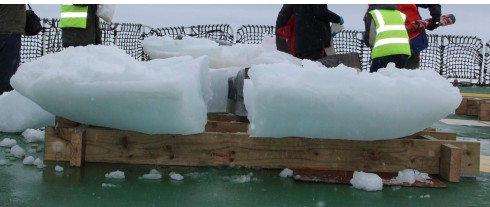

**Figure 5.** Pancake ice floes placed on wooden grids on the ship's helideck: Pancake A (top left), Pancake B (top right), Pancake C (bottom left) and Pancake D (bottom right) shattered into six large pieces during removal from the net.

**Table 2.** Overview of pancake ice floe dimensions.

| Pancake ice floe | | A | B | C | D |
|---|---|---|---|---|---|
| | Height | 0.40 | 0.37 | 0.45 | 0.37 |
| Dimensions [m] | Length | 2.42 | 2.49 | 2.40 | 2.95 |
| | Width | 1.83 | 2.33 | 2.33 | 2.51 |

All four pancakes had rafted edges, were snow-covered and did not have a visibly discoloured biologically active layer. The depth and temperature of the snow-layer was taken before each pancake was manoeuvred out of the net which is found in Tab. 2. During the pancake ice floe lifting, the 3D printed Niskin bottle was again deployed from the helideck to retrieve water samples from which the ice-ocean boundary layer salinity was determined. In addition, interstitial frazil ice was collected and its viscosity was measured, further details are found in Paul et al. (2021).

Subsequently, between 17 and 23 cores were obtained from each pancake ice floe. The corresponding coring layout with core IDs and corresponding names can be found in App. A2 in Fig. A1 and Tab. A2-A5, respectively. As for the pack ice, all cores were extracted in vertical direction, perpendicular to the ice floe surface. As such, the length of the cores is representative of the pancake ice floe thickness (depth). For each floe, three cores were tested to determine the vertical temperature profile immediately after the floe had been off-loaded. All cores were subsequently transferred to either the Cold Laboratory for further

processing and testing, or to the ship's freezer.

## 2.2 Testing of physical and mechanical properties

The ice cores used to profile the vertical temperature distribution in consolidated pack ice and pancake ice were subsequently cut with a stainless steel band saw into segments close to $10\,\mathrm{cm}$-length and allowed to melt in ziploc bags in a refrigerator at $5\,°\mathrm{C}$. Once melted, the volume of the samples was measured using a graduated beaker and the bulk salinity was obtained by means of a 8410A Portasal salinometer at room temperature. On deck sampled pancake ice was processed within $30\,\mathrm{min}$ and overboard sampled consolidated pack ice within $2.5\,\mathrm{h}$ after core extraction.

As the working conditions in the Cold Laboratory aboard the ship were challenging due to the significant swell at edge of the Antarctic MIZ, it was decided to compute the brine volume distributions in pancake ice and consolidated pack ice only based on the obtained temperature and salinity data via the empirical relations by Frankenstein and Garner (1967)

$$V_b = S_i \left( \frac{49.185}{T} + 0.532 \right) \tag{1}$$

where $V_b$ denotes brine volume [‰], $S_{si}$ [PSU] the bulk salinity and $T$ [°C] the absolute value of the temperature. The more accurate relation by Cox and Weeks (1983) additionally requires accurate enough ice density measurements which were difficult to obtain aboard the ship.

The sampled pancake and consolidated pack ice was studied with respect to crystal structure, stratigraphy and texture. For this purpose, one ice core at consolidated conditions and further four pancake ice cores were investigated. The ice cores were visually inspected already during extraction to record obvious physical features such as thickness of snow topping, layering in the texture of the ice, and the presence of possible defects such as cracks. The cores were then taken directly to the ship's storage facility and analysed back at the University of Cape Town, approximately 6 months after collection being stored at $-20\,°\mathrm{C}$. The cores were first cut into $10\,\mathrm{cm}$-segments and longitudinal sections with thicknesses in the order of $8\,\mathrm{mm}$ were then cut from the centre of these segments with a stainless steel band saw at $-10\,°\mathrm{C}$ laboratory temperature. These $8\,\mathrm{mm}$-thick sections were further cut using a thermal macrotome to create thin sections of $1\,\mathrm{mm}$ thickness and subsequently viewed through cross-polarised sheets and photographed. The used thermal macrotome was constructed by the authors to produce thin sections for cross-polarisation viewing. This device uses the concept of heat being passed through a nickel-chrome wire that slices a section of sea ice. This device is able to cut ice core sections to a thickness of $1\,\mathrm{mm}$ or less. The cross-polarised photographs were subsequently analysed for their respective ice textures and crystal sizes. From these images, stratigraphy diagrams were composed for easy visual comparison between cores and the average crystal sizes measured using the scale on the photographs. An average of 20 random crystals per texture layer were selected for measurement in a photograph and the average size and standard deviation was calculated. Furthermore, a 95-percentile confidence interval test was performed over the data collected for each texture for statistical comparison.

The testing of anisotropic elastic material properties of sea ice was conducted in the ship's Cold Laboratory at $-10\,°\mathrm{C}$ within a few hours after collection. Since the mechanical behaviour of sea ice is rate-dependent, the dynamic Young's modulus is typically determined measuring the speed of ultrasonic waves, a pressure and shear wave, travelling through the ice to exclude viscous strain in the deformation response (Timco and Weeks, 2010). The P- and S-wave signals are initiated by a pair of transducers put on opposite surfaces of a specimen. The Young's and shear moduli can then be calculated on the basis of

its length (the distance between the transducers), its bulk density as well as the speed of transmission of the P- and S-waves in accordance with the following formulae (Snyder et al., 2015):

$$E_i = \rho c_s^2 \frac{3c_p^2 - 4c_s^2}{c_p^2 - c_s^2}, \tag{2}$$

$$G_i = \rho c_s^2, \tag{3}$$

where $E_i$ and $G_i$ denote the dynamic Young's and shear moduli, respectively, associated with the material direction $x_i$, $\rho$ the bulk density of ice, $c_p$ the transmission speed of the P-wave and $c_s$ the transmission speed of the S-wave, both measured along direction $x_i$.

The $9\,$cm-diameter cores ranging between 30-45 cm in length were first visually inspected in the laboratory to identify any internal cracking or other defects that could affect the ultrasound testing results. Subsequently, it was planned how to segment the cores into specimen lengths as close to $10\,$cm as possible for testing, see Fig. B1 in App. B. At this stage, consideration was given to already existing segmentation due to fracturing occurred at time of core extraction. The cores were then cut into the planned segments along the entire length using a stainless steel bandsaw providing clean parallel surfaces at the end faces on either side for the subsequent ultrasound measurements in the longitudinal ice core direction. For measurements in the transverse core direction, each segment was again cut in longitudinal direction at two parallel opposite sides providing the required flat surfaces to apply the transducers. These two surfaces were $7.2\,$cm $\pm\,0.8\,$cm and $7.1\,$cm $\pm\,1.3\,$cm apart from each other for pancake and consolidated pack ice specimens, respectively. In order to determine the bulk density of the 43 specimens, their dimensions and weight were first measured in the Cold Laboratory at $-10\,°$C, the latter using a hand-held scale. The ultrasound testing was carried out less than two hours after core extraction using a PROCEQ Pundit PL-200 testing kit (Screening Eagle Technologies, Schwerzenbach, Switzerland with $250\,$kHz P- and S-wave transducers assuming that the mechanical properties of the ice cores had not significantly changed by that time (Pustogvar and Kulyakhtin, 2016). A low-temperature silicon grease applied to the transducer faces ensured the best possible contact with the ice specimens for the transfer of P- and S-waves across the interface. Three separate tests were carried out for each specimen and direction.

The compressive strengths of the ice samples were determined using the GCTS PLT-2W wireless unconfined compression testing device (GCTS Testing Systems, Tempe, USA). It has a virtually infinite stiffness by automatically correcting for the frame deformation based on the location of the top cross-head and as a function of the axial load. Samples with a height up to $13.5\,$cm can be accommodated. There are no constraints induced by the platens to avoid a triaxial state of stress. The cores were cut via a stainless steel bandsaw into multiple $13.5\,$cm-long segments and immediately tested. In the interest of time to minimize brine drainage and alterations to ice composition, it was deliberately decided not to reduce the sample's diameter required for the optimal diameter/length-ratio of 1:2.5 (Schulson and Duval, 2009). In total, 27 compression testing specimens were obtained from twelve pancake ice and three pack ice cores. Specimen dimensions and names are displayed in App. B in Fig. B2. The specimen naming convention is derived from the core names with an additional numbered postfix (S1-S6) indicating the consecutive specimen segmentation sequence starting from the top of the core. The compression testing commenced less than two hours after core extraction. The uniaxial compression load was manually applied to the sample by a hand pump with an approximate strain rate of $2 \times 10^{-4}\,\mathrm{s}^{-1}$ to $6 \times 10^{-4}\,\mathrm{s}^{-1}$.

An estimate of error in the measurement and registration of temperature, salinity, density, elastic properties and compression strength is less than 9%, which is each significantly less than errors introduced by the variation of the ice structure and properties.

## 3 Results and discussion

### 3.1 Temperature, salinity, porosity and density

The vertical temperature profiles for consolidated pack ice and pancake ice floes A, B and D illustrated in Fig. 6 exhibit the typical linear distribution with depth (Nakawo and Sinha, 1981; Petrich and Eicken, 2017). The temperature gradient of the former is considerably larger due to the lower atmospheric temperature on the day of sampling and testing. However, for pancake ice floes C, the temperature distribution is nearly constant and rather high temperature values were recorded, in particular toward the top core sections. Accordingly, its consistency was reported to be rather soft and poorly consolidated except for the very top. A significant influence of the snow cover on this disparity can be ruled out, as only minor snow cover was found as listed in Tab. 3 and shows only small variations between all four pancakes. The substantially increased temperatures and porous texture of pancake ice floe C are exhibited in particular in top and middle core sections, a potential explanation could therefore be overwashing occurred during collection and subsequent vertical drainage. This is also further substantiated by the temperature measurements of the thin snow layers covering the four pancake ice floes performed on deck providing $-2.8 \pm 0.1\,°C$ which indicates overwashing as it should be closer to the measured atmospheric temperature of $-8\,°C$. Furthermore, the snow temperatures do not link to their respective pancake ice floe temperature profiles, except of pancake ice floe C. It has been reported from ice floe-mounted SIMBA measurements of storm-caused ice floe flooding events that subsequent significant heat propagation occurred increasing ice temperature from initially $-8.5\,°C$ at the top surface by about $3\,°C$ down to $70\,cm$ depth (Provost et al., 2017).

The boundary layer salinity of the ocean was determined as 33.29 and 34.11 PSU at MIZ3A and MIZ1D, respectively. The vertical salinity profile for the pancake ice floes depicted in Fig. 7 exhibits at the top of pancake ice floes with $12.6 \pm 4.9\,PSU$ significantly higher values than below with $5.8 \pm 1.1\,PSU$. Comparable characteristics in terms of depth evolution and magnitude of salinity have been reported in the Weddell Sea by Eicken (1992), Doble et al. (2003) and Tison et al. (2017). The high salinity of the topmost core section is generally attributed to the high initial growth rate and lower permeability of fine-grained granular ice (Nakawo and Sinha, 1981). This can be confirmed by the stratigraphy of the cores of pancake ice floes A, B and C which are entirely composed of granular ice as shown in Fig. 10. The obtained values compare to in situ data of congealed grease ice in an artificial lead by Smedsrud and Skogseth (2006). On the other hand, in situ experiments in the Arctic by Notz and Worster (2008) demonstrated that initially high salinity values at the topmost parts giving the characteristic C-shape vanish due to gravity drainage within $48\,hrs$, if the frazil ice layer is thin. This would explain the generally low salinity values at the top of the sampled consolidated pack ice of 4.4-7.4 PSU where the topmost granular layer is only about $5\,cm$-thick as shown in Fig. 10 and is immediately followed by a columnar layer.

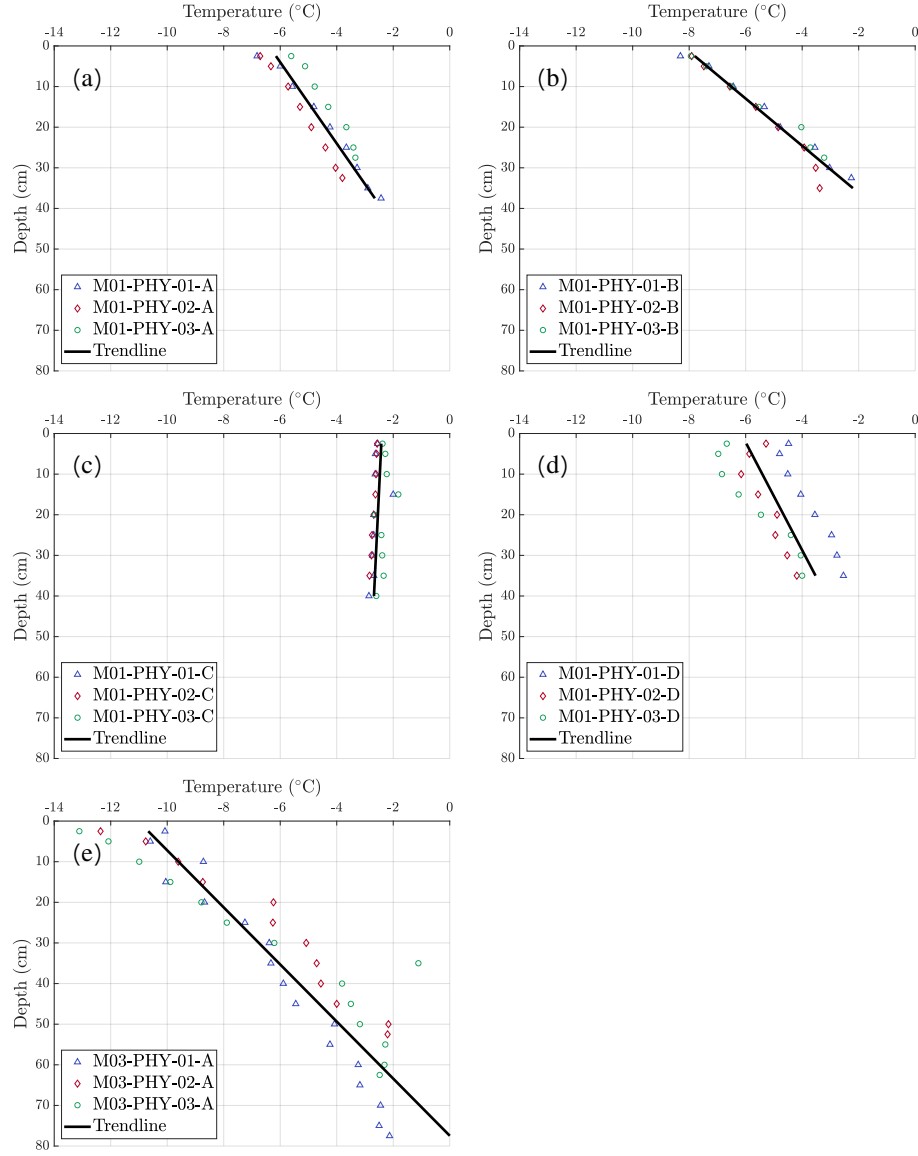

**Figure 6.** Plots of temperature over sea ice depth for the four pancake ice floes (cores M01-PHY-01-A/B/C/D, M01-PHY-02-A/B/C/D, M01-PHY-03-A/B/C/D) and for the consolidated pack ice (cores M03-PHY-01-A, M03-PHY-02-A and M03-PHY-03-A), respectively.

The snow depth and salinity measurements of the pancake ice floes is listed in Tab. 3. The snow salinity is noticeably higher than the salinity found at the topmost core sections of pancake ice floe A, B and D pointing at surface flooding which was also found for East-Antarctic sea ice (Massom et al., 1998). For floe C the salinity measurements in snow and underlying ice are of a similar magnitude. The latter might be connected to significant overwashing and drainage occurred during floe collection as
mentioned before.

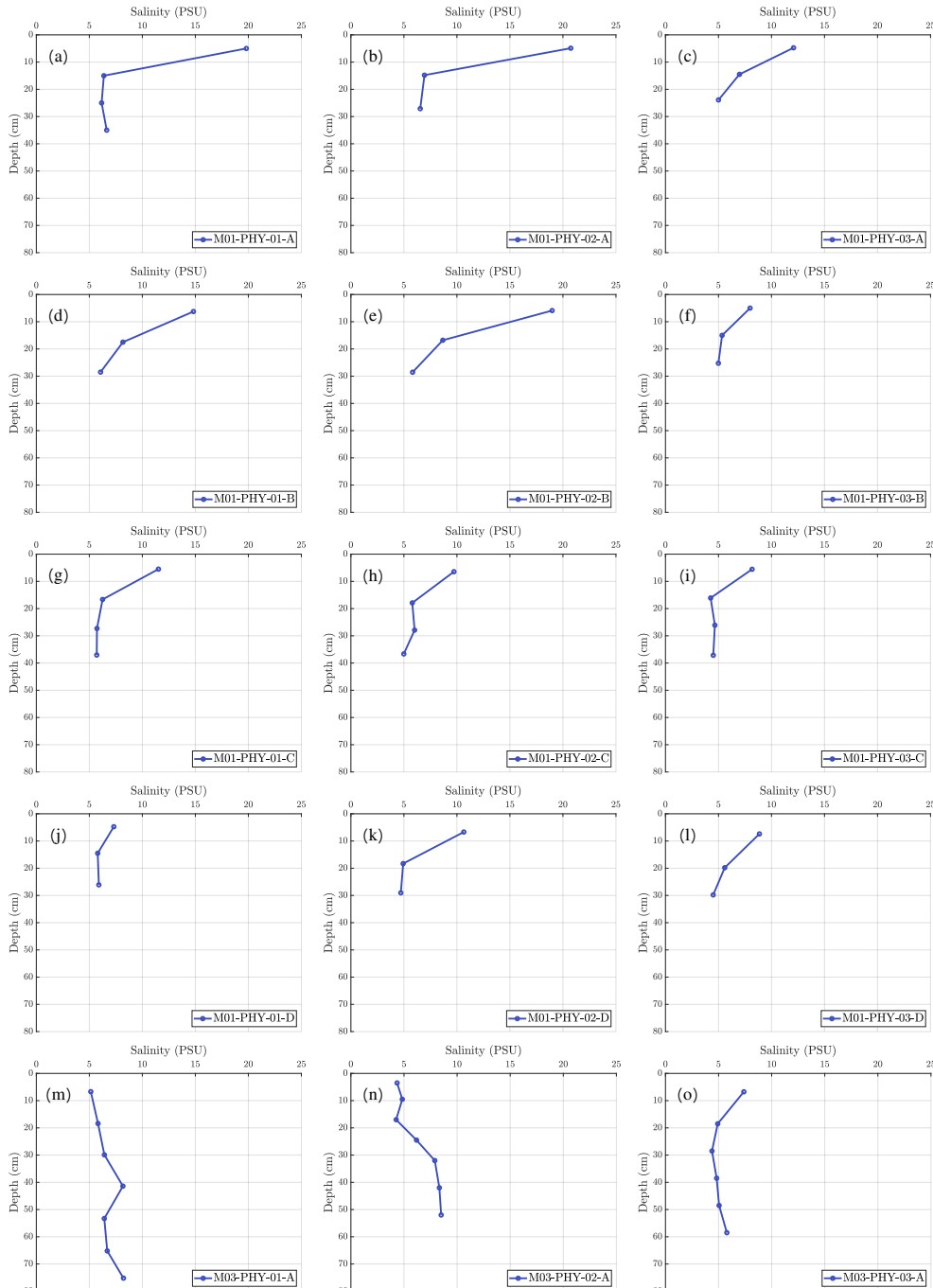

**Figure 7.** Plots of salinity over sea ice depth for the four pancake ice floes (cores M01-PHY-01-A/B/C/D, M01-PHY-02-A/B/C/D, M01-PHY-03-A/B/C/D) and for the consolidated pack ice (cores M03-PHY-01-A, M03-PHY-02-A and M03-PHY-03-A), respectively.

**Table 3.** Overview of pancake ice floe snow depth and salinity.

| Pancake ice floe | A | B | C | D |
|---|---|---|---|---|
| Snow depth [cm] | 2.5 | 3.5 | 3.6 | 3.5 |
| Snow salinity [PSU] | 24.74 | 18.43 | 9.68 | 14.26 |

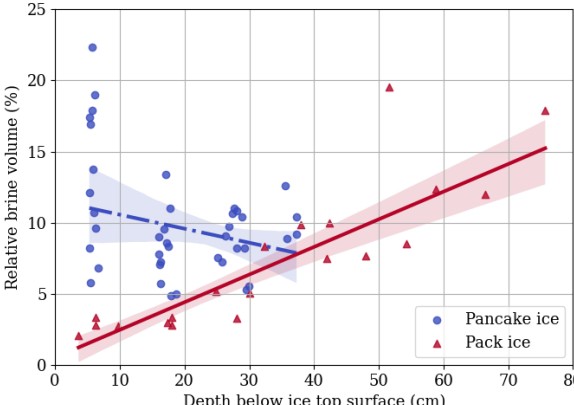

**Figure 8.** Brine volume as a percentage of the total volume as a function of depth for pancake ice floes and consolidated pack ice at stations, MIZ1D and MIZ3A, respectively. The shading indicates the 90-percentile confidence intervals.

Due to brine drainage, the computed brine volume in consolidated pack ice is generally lower than in pancake ice (Notz and Worster, 2008) which is confirmed by the computed brine volume data illustrated in Fig. 8. The pancake ice brine volumes obtained are above the permeability threshold of 5% which also has been found by Tison et al. (2017) in the MIZ of the Weddell Sea. As the measured salinity in consolidated pack ice is fairly constant with $6.0 \pm 1.5$ PSU, the brine volume increases with temperature and depth ($r = 0.866$, $n = 20$, $p < 0.001$). The opposite is found for pancake ice where the brine volume clearly decreases with depth ($r = -0.379$, $n = 40$, $p = 0.016$), as the salinity values at the topmost sections are significantly higher compared to the lower sections. However, it should be noted that significant brine loss has been observed during sampling, in particular, when the pancake ice floes were lifted out of the ocean and placed on deck of the ship. As such the actual salinity and relative brine volume of the sampled pancake ice would be higher and the actual bulk density lower than reported in Figs. 7, 8 and 9, respectively, in particular for the bottom of the ice. This aspect has also been observed by Eicken (1992). Using the empirical relation by Frankenstein and Garner (1967) (Eq. (1)), only the uncertainty of salinity measurements negatively impact on the accuracy of the relative brine volume computation, as the temperature measurements reflect in situ conditions.

Determining sea ice density in the field aboard a ship is generally less accurate than under controlled laboratory conditions on land due to a higher degree of measuring errors with regards to specimen dimensions and weight. Considering that the density values for pancake ice illustrated in Fig. 9 are with $879 \pm 53.5$ kg m$^{-3}$ noticeably lower than that of pure ice, a significant air volume fraction is indicated which can be partly attributed to prior brine loss occurred during ice sampling as previously

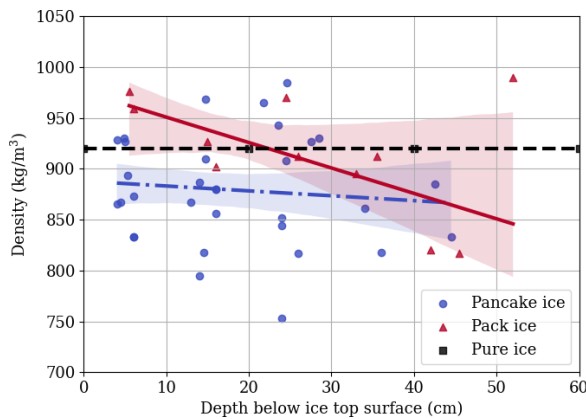

**Figure 9.** Bulk density as a function of depth for pancake ice floes and consolidated pack ice at stations, MIZ1D (cores M01-US-01A/B/C/D, M01-US-02A/B/C/D) and MIZ3A (cores M03-US-01-A and M03-US-02-A), respectively. The shading indicates the 90-percentile confidence intervals.

mentioned. For the consolidated pack ice, measurement errors and small sample size in combination yield a bulk density trendline which is not plausible for the top parts, as it would correspond to very high salinity values which have not been observed. There is no significant correlation between density and depth for pancake ice ($r = -0.103$, $n = 32$, $p = 0.576$) and pack ice ($r = -0.426$, $n = 11$, $p = 0.192$).

## 3.2 Texture and fabrics

Three different ice textures can be identified in the cores, namely granular, transitional and columnar. Granular textures, associated with the initially formed frazil ice and also meteoric ice (Massom et al., 1998), have the smallest crystal grain sizes (Eicken, 1998). Granular ice of meteoric origin is due to snow deposited on pancake ice floes mixing with ocean water through percolation for sufficiently heavy snow loading or overwashing, the latter is a common phenomenon in the Ross, Amundsen, Bellingshausen and Weddell Seas (Eicken et al., 1995; Jeffries et al., 1994; Sturm et al., 1998). The proportion of snow ice is significant in particular in Antarctic sea ice due to abundance of precipitation and severe sea states (Bromwich et al., 2004; Jeffries et al., 2001; Lange et al., 1990; Worby et al., 1998). Columnar ice textures display large, elongated crystals while transitional textures display a mixture of granular and columnar textures, thus its defining features are small crystals that are beginning to elongate and lengthen into the distinctly columnar ice.

A summary of the fraction of ice textures found to be present in each core along with their respective crystal sizes is found in Tab. 4, while the stratigraphy diagrams are illustrated in Fig. 10. The stratigraphy diagrams are a summary showing the vertical layering of the ice textures within the sampled sea ice. The pancake ice floes A, B and C are completely comprised of granular ice as shown in Fig. 10 which is expected of young pancake ice found at the edge of the Antarctic marginal ice zone where frazil ice generation is prevalent due to the harsh sea states in winter (Lange and Eicken, 1991). It is known from in situ

**Table 4.** Summary of the sea ice core textures. $\mu$ specifies the 95-percentile confidence intervals.

| Station-Floe | Granular | | Transitional | | Columnar | |
| --- | --- | --- | --- | --- | --- | --- |
| | Fraction | Crystal Size [mm] | Fraction | Crystal Size [mm] | Fraction | Crystal Size [mm] |
| MIZ1D-A | 1 | 2.06 | - | - | - | - |
| MIZ1D-B | 1 | 2.31 | - | - | - | - |
| MIZ1D-C | 1 | 1.77 | - | - | - | - |
| MIZ1D-D | 0.5 | 1.38 | 0.5 | 4.34 | - | - |
| MIZ3A | 0.84 | 2.58 | 0.067 | 7.55 | 0.093 | 15.6 |
| | $1.87 < \mu < 2.17$ | | $4.77 < \mu < 6.23$ | | - | |

observations in the Arctic that the grease ice layer thickness can reach more than $40\,\mathrm{cm}$ before congealing under turbulent and dynamic conditions characterized by convective overturning and wind-driven frazil accumulation (Smedsrud and Skogseth, 2006). Pancake ice floe D also displays transitional ice textures indicating the beginning of downward freezing due to calmer

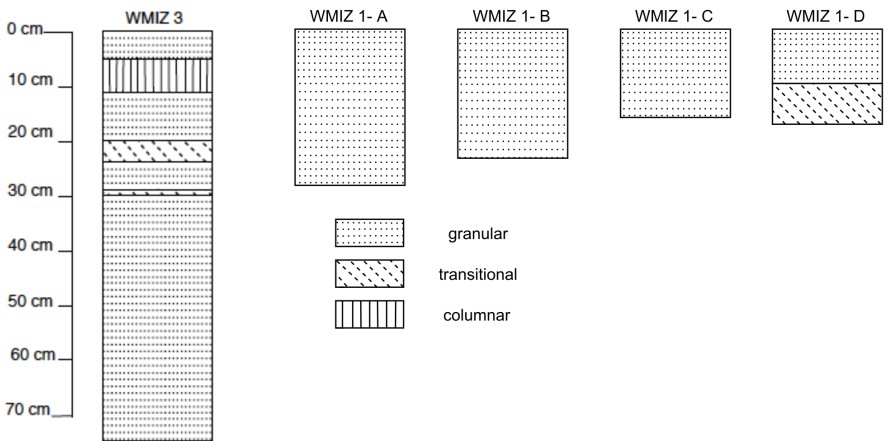

**Figure 10.** Stratigraphy diagrams for consolidated pack ice core M03-CP-01-A (WMIZ 3) and pancake ice cores M01-CP-01-A (WMIZ 1-A), M01-CP-01-B (WMIZ 1-B), M01-CP-01-C (WMIZ 1-C) and M01-CP-01-D (WMIZ 1-D), respectively.

sea state and wind conditions (Weeks and Ackley, 1982). The variability in the pancake ice floes is hard to quantify with
285 regards to their internal composition due to their young age. The size range of the granular crystals varies, with pancake ice floe A and C being statistically dissimilar.

The consolidated pack ice core clearly indicates the presence of different layers and is found to be made up of 84 % granular crystals. Predominantly granular sea ice textures are characteristic for the Weddell Sea region as reported in literature (see

e.g. Lange and Eicken (1991); Tison et al. (2017)). As previously mentioned, this is generally indicative of fairly turbulent ice growth conditions leading to significant frazil ice generation. Small bands of columnar and transitional ice textures were found sandwiched between granular ice which is typical of young ice grown in a highly dynamic ocean environment where regularly occurring atmospheric storms can interrupt steady growth of the ice, causing ice crystals and grain structures to vary in type, size and orientation during the growth period of the ice (Lange and Eicken, 1991; Carnat et al., 2013; Shokr and Sinha, 2015).

### 3.3 Elastic properties

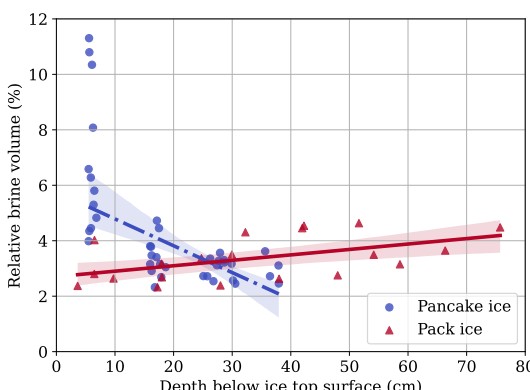

**Figure 11.** Brine volume for uniform ice temperature of $-10\,^{\circ}\mathrm{C}$ as a percentage of the total volume as a function of depth for pancake ice floes ($r = -0.658$, $n = 40$, $p < 0.001$) and consolidated pack ice ($r = 0.508$, $n = 20$, $p = 0.022$) at stations, MIZ1D and MIZ3A, respectively. The shading indicates the 90-percentile confidence intervals.

In order to allow for the direct comparison of ice stiffness and strength with relative brine volume, the latter is recomputed via Eq. (1) using the actual ice temperature of constant $-10\,^{\circ}\mathrm{C}$ when being processed in the Cold Laboratory instead of the in situ-measured temperature distribution. The resulting relative brine distribution with depth is depicted in Fig. 11. The uniform temperature distribution over depth does not affect the general trend of porosity over depth when compared with Fig. 8. However, it yields noticeably lower porosity values for pancake ice at the top and for consolidated pack ice at the bottom.

The vertical distributions of dynamic Young's and shear moduli measurements over depth for consolidated pack ice and pancake ice are shown in Figs. 12 and 13, respectively. For the pancake ice, there is no significant correlation between Young's modulus and depth in longitudinal direction ($r = 0.120$, $n = 32$, $p = 0.513$) and in transverse direction ($r = 0.078$, $n = 32$, $p = 0.673$). Equally, there is also no significant positive correlation between shear modulus and depth in longitudinal direction $r = 0.115$, $n = 32$, $p = 0.531$) and in transverse direction ($r = 0.045$, $n = 32$, $p = 0.809$). For consolidated pack ice, there is no significant negative correlation between Young's modulus and depth in longitudinal direction ($r = -0.413$, $n = 11$, $p = 0.207$), however, there is a significant correlation in transverse direction ($r = -0.672$, $n = 11$, $p = 0.024$). Similarly, there is no significant correlation between shear modulus and depth in longitudinal direction ($r = -0.425$, $n = 11$, $p = 0.192$), but

there is a significant correlation in transverse direction ($r = -0.665$, $n = 11$, $p = 0.026$). As the brine volume is increasing with depth for consolidated pack ice as shown in Fig. 11, this correlation is in-line what has been reported in literature, see e.g. Langleben and Pounder (1963); Mellor (1986); Moslet (2007).

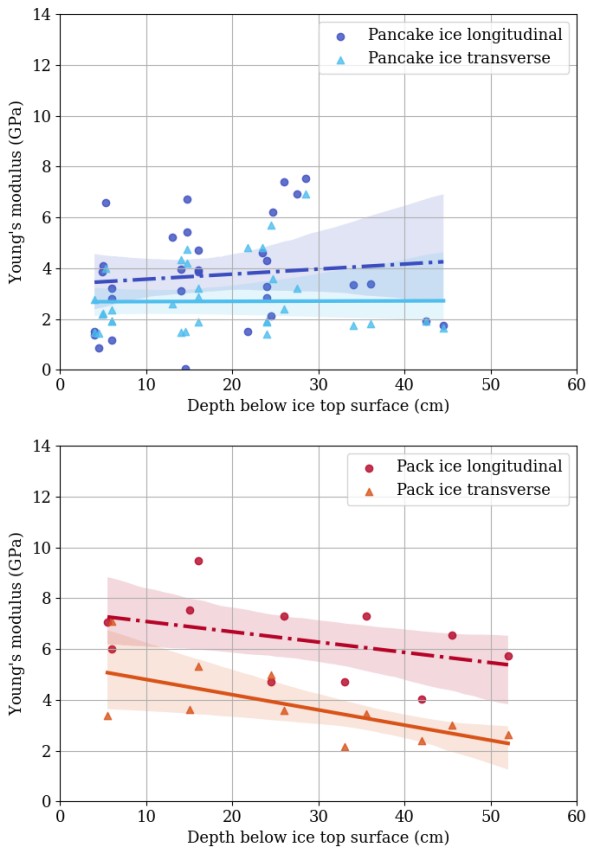

**Figure 12.** Longitudinal and transverse Young's modulus for pancake ice cores M01-US-01-A/B/C/D and M01-US-02-A/B/C/D as well as consolidated pack ice cores M03-US-01-A and M03-US-02-A as a function of depth from the top of the ice. The shading indicates the 90-percentile confidence intervals.

With substantially lower brine values as shown in Fig. 11, the elastic properties found for consolidated pack ice exhibit higher values than for pancake ice, in particular for the upper core sections, with values in longitudinal direction of $6.4 \pm 1.6\,\mathrm{GPa}$ and $2.3 \pm 0.6\,\mathrm{GPa}$ for the Young's and the shear modulus, respectively, as opposed to $3.7 \pm 2.0\,\mathrm{GPa}$ and $1.3 \pm 0.7\,\mathrm{GPa}$, respectively. Also, the textural differences between both ice types can be expected to contribute to this discrep-
315 ancy. The stratigraphical data found for consolidated pack ice (Fig. 10) indicates granular layers intermixed with transitional and columnar layers located in the upper half of the core whereas pancake ice is almost exclusively granular. The dynamic Young's and shear moduli for both ice types are generally lower than reported in literature for columnar first-year ice at low

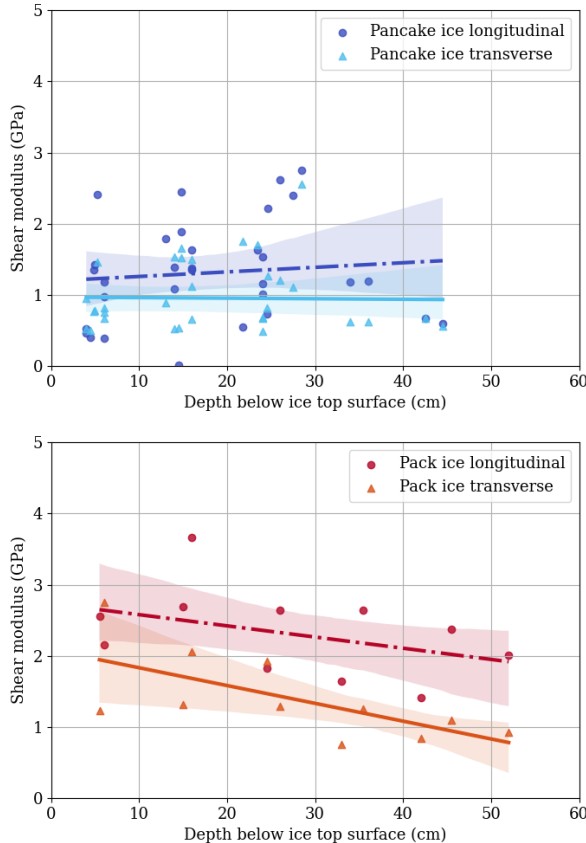

**Figure 13.** Longitudinal and transverse shear modulus for pancake ice cores M01-US-01-A/B/C/D and M01-US-02-A/B/C/D as well as consolidated pack ice cores M03-US-01-A and M03-US-02-A as a function of depth from the top of the ice. The shading indicates the 90-percentile confidence intervals.

brine volumes given as 9.0-9.5 GPa (Mellor, 1986) and 3.0-3.5 GPa (Snyder et al., 2015), respectively. For the consolidated pack ice, this is most likely due to the high granular ice content as temperature and salinity values are comparable.

In the sea ice studies by Sinha (1984) and Nanthikesan and Sunder (1994) based on the experiments by Dantl (1969) as well as Gammon et al. (1983), polycrystalline sea ice is found to exhibit only a very slight directional dependency in terms of elastic material properties. In contrast, the results shown in Figs. 12 and 13 indicate a significant difference of elastic moduli measured in longitudinal and transverse core directions, respectively, in particular for the consolidated pack ice where the latter was determined as $3.8\pm1.5$ GPa and $1.4\pm0.6$ GPa for the Young's and the shear modulus, respectively. Based on the previous

studies mentioned before, the finding of directional dependence of elastic properties in this magnitude is unprecedented, in particular considering the predominantly granular ice composition. The pore and channel structure might be of influence which, for consolidated pack ice, can be expected to increase with depth leading to the formation of predominantly vertically aligned brine channels which potentially alter the directional dependency of elastic ice properties with depth.

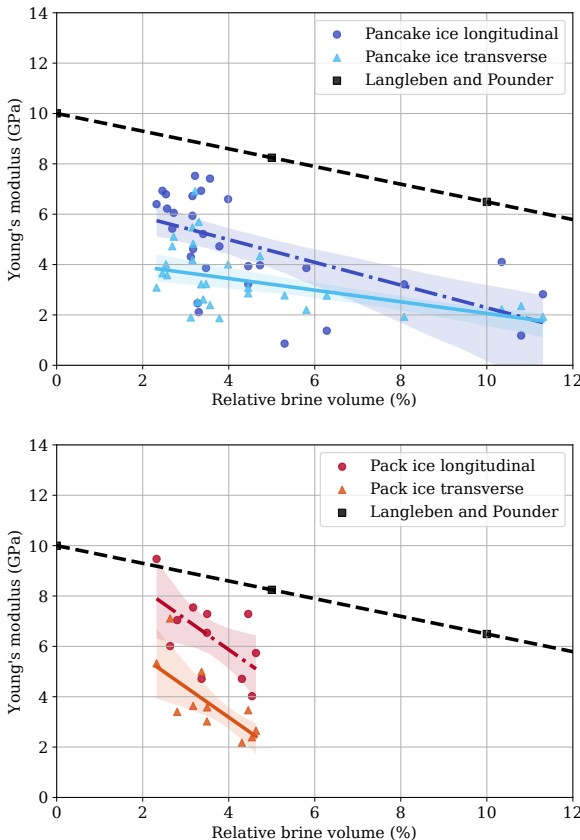

**Figure 14.** Longitudinal and transverse Young's modulus for pancake ice cores M01-US-01-A/B/C/D and M01-US-02-A/B/C/D as well as consolidated pack ice cores M03-US-01-A and M03-US-02-A as a function of percentage porosity of total volume. The shading indicates the 90-percentile confidence intervals.

Combining brine volume and Young's modulus datasets presented in Figs. 11 12 and 13, respectively, the dependency of the Young's modulus on the relative brine volume is obtained as shown in Fig. 14. There is a significant correlation between Young's modulus and brine volume for pancake ice in longitudinal direction ($r = -0.579$, $n = 29$, $p = 0.001$) and in transverse direction ($r = -0.493$, $n = 29$, $p = 0.007$). For consolidated pack ice, there is also a significant correlation between Young's modulus and brine volume in longitudinal direction ($r = -0.597$, $n = 11$, $p = 0.053$) and in transverse direction ($r = -0.741$, $n = 11$, $p = 0.009$). A remarkable aspect of the obtained data is the noticeable directional dependency of the stiffness-porosity relation, as the Young's modulus, in particular in transverse direction, follows distinctly lower trend-lines in terms of magnitude than predicted by the relation found by Langleben and Pounder (1963) for Arctic columnar ice indicated by the dashed black line in Fig. 14 which is expressed as

$$E(\text{GPa}) = 10.0 - 35.1 V_b$$

where $V_b$ [%] denotes the brine volume. A closer match with the curve by Langleben and Pounder could be expected for the stiffness-porosity data in longitudinal direction for the pancake floe ice, if no brine loss had occurred during sampling. However, the brine loss for the consolidated pack ice was less substantial. As such textural differences between the sampled predominantly granular Antarctic sea ice and the columnar Arctic sea ice appear to be of significance.

## 3.4 Compressive strength

Compressive strength of Antarctic sea ice, specifically in winter, has only been studied by Urabe and Inoue (1988a), but the testing was done after long storage-time at $-20\,°\mathrm{C}$. In contrast, the compressive stress-time plots for the fifteen cores displayed in Fig. 15 have been obtained in situ within two hours of collection. The uniaxial compressive strength for pancake ice and consolidated pack ice were determined as $2.3\pm0.5\,\mathrm{MPa}$ and $4.1\pm0.9\,\mathrm{MPa}$, respectively. All tests shown in Fig. 15 are characterized by ductile failure behaviour with initial strain hardening and significant strain softening after the peak stress has been reached. For columnar ice, Kuehn and Schulson (1994) found the ductile-brittle failure transition at a strain rate of $2\times10^{-4}\,\mathrm{s}^{-1}$. However, it was stated to be one order of magnitude higher for granular ice which can be confirmed by our results obtained for a strain rate of $5\times10^{-4}\,\mathrm{s}^{-1}$.

The Antarctic sea ice compressive strength results by Urabe and Inoue (1988a) of about $8.5\,\mathrm{MPa}$ for vertical tested samples at a strain rate of around $1\times10^{-4}\,\mathrm{s}^{-1}$ are higher than our strength measurements for consolidated pack ice. However, they tested land-fast sea ice. The compression strength data by Kivimaa and Kosloff (1994a) obtained in the Weddell Sea in situ in spring are in the range from $1.2\,\mathrm{MPa}$ to $4.5\,\mathrm{MPa}$ at a strain rate of $1\times10^{-3}\,\mathrm{s}^{-1}$, which, despite the brittle failure regime, is lower than the consolidated pack ice data obtained in this study.

It is known from literature, that an increase in ice temperature results in a lower compressive strength due to increasing porosity (Han et al., 2015; Kermani et al., 2007; Moslet, 2007). This linkage can be confirmed for the tested consolidated pack ice considering the depth-evolution of both, the increasing relative brine volume illustrated in Fig. 11 and the decreasing maximum stress ($r=-0.632$, $n=8$, $p=0.093$) depicted in Fig. 16. Similarly, for the tested pancake ice, the relative brine volume decreases with depth whereas the maximum uniaxial compressive stress increases ($r=-0.452$, $n=19$, $p=0.052$). The increase of pancake ice compressive strength over depth can be fitted by the following linear trend-line

$$\sigma_{\mathrm{compression}} = (4.4\cdot d+1.6)\,\mathrm{MPa} \tag{4}$$

where $d$ denotes the depth in meter. This discrepancy between the tested consolidated pack and pancake ice can be explained by the high salinity values and brine volumes at the top of the pancake ice floes which show the opposite trend compared with the consolidated pack ice as illustrated in Fig. 11 . Kovacs (1997) proposed a relation based on total porosity and strain rate to estimate the compressive strength of Arctic first-year sea ice. Fig. 17 compares this relation with the measured compressive strength of pancake ice and consolidated pack ice making use of the porosity data given in Fig. 11. The compressive strength clearly shows lower magnitudes than expected by the relation derived by Kovacs, in particular for the pancake ice samples. This can be partly contributed to the previously mentioned brine loss at sampling which has been more significant for the

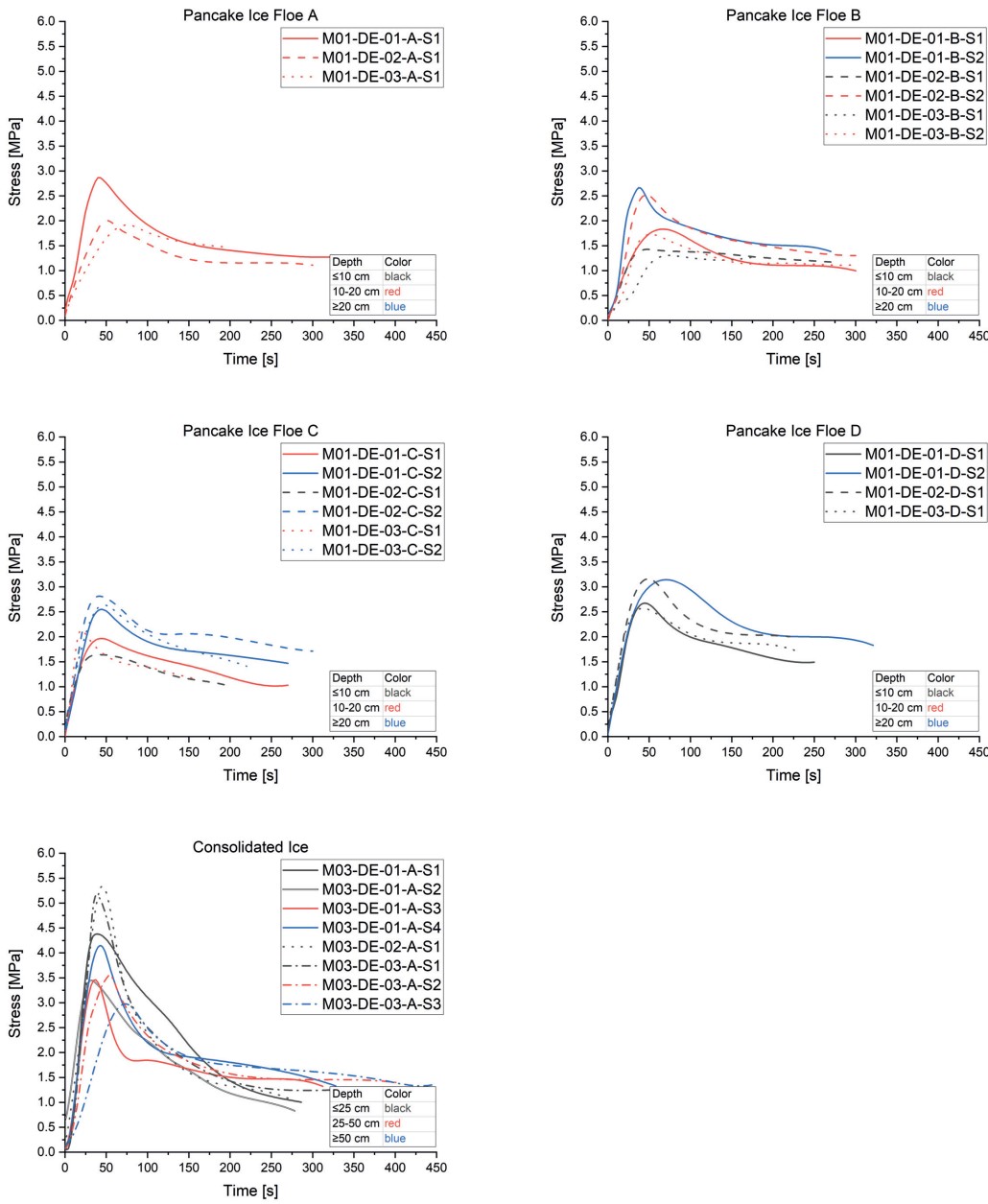

**Figure 15.** Stress-time plots of uniaxial compression test results for all specimens, respective pancake ice floes and consolidated pack ice. The color-coding refers to the sample depth in cm along the core length measured at the sample center and the line style to the specific core.

pancake ice floes than for consolidated pack ice. However, differences between Arctic and Antarctic ice compositions in terms of texture and fabrics might also play a role warranting further studies in this regard.

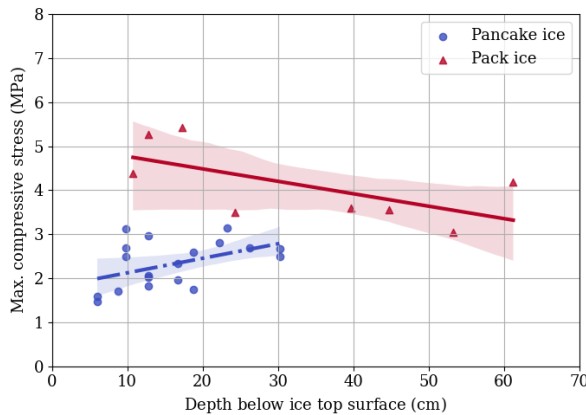

**Figure 16.** Maximum uniaxial compressive stress for pancake and pack ice over depth with the 90-percentile confidence intervals indicated.

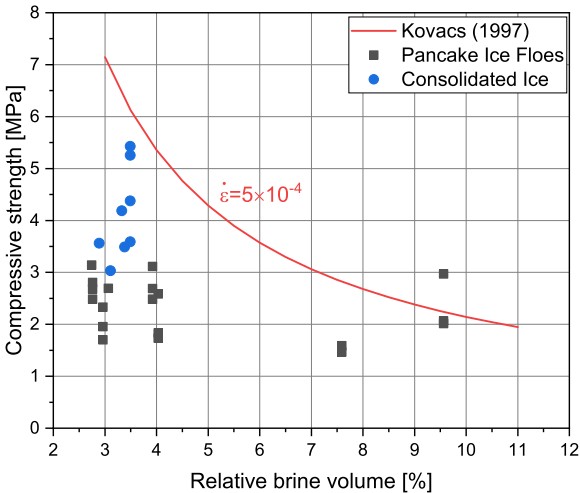

**Figure 17.** Uniaxial compressive strength for pancake ice and consolidated ice against brine volume as compared to the relation found by Kovacs using the strain rate of $5 \times 10^{-4}\,\mathrm{s}^{-1}$ applied to obtain the here reported data.

Sinha (1984) reported an increasing ice strength with depth in horizontally tested hummock ice of an old floe. Kivimaa and Kosloff (1994b) as well as Urabe and Inoue (1988a) found that the maximum compressive strength was reached in a middle layer of the ice sheet. In the study by Frederking and Timco (1983), a difference was also observed between top and bottom layers where the latter exhibits a larger strength magnitude. This is in contrast to our findings regarding consolidated pack ice which is strongest at the top and exhibits decreasing strength towards the middle layer due to the increasing brine volume with depth.

Both, consolidated pack and pancake ice investigated in this study, are relatively young first-year ice which has been collected in the Antarctic MIZ during the winter season when sea ice was advancing and was exposed to high temperature gradients. In this sense, its measured strength properties are therefore not directly comparable to those mentioned before.

## 4 Conclusions

The collected data sets of young first-year ice combined physical, textural and mechanical in situ test results of solitary pancake ice floes and consolidated pack ice representing a vertical $150\,\text{km}$-transect into the advancing Antarctic marginal ice zone along the Good Hope Line. Of particular interest was to elucidate the transition between both ice types within the MIZ in terms of their differences in mechanical stiffness and strength properties as linked to physical and textural characteristics at early-stage ice formation.

Both ice types exhibited a characteristic linear temperature profile with depth and had predominantly granular textures, for pancake ice almost exclusively, due to the highly dynamic ocean environment. Pancake ice floe C, however, had an almost constant and fairly high temperature distribution which was attributed to overwashing during collection and subsequent vertical drainage. The pancake ice salinity profile exhibited very high magnitudes at the topmost sections and in the snow cover. Accordingly, overwashing and flooding events are a reasonable explanation for the generally high pancake ice salinity at the topmost sections, in particular, as the consolidated pack ice had a very low salinity value at the top, slightly increasing toward the bottom, where recent flooding events can be excluded. Due to the relatively high porosity and brine loss occurred during sampling of pancake floe ice, the recorded bulk salinity, relative brine volume and density can be assumed to be lower than in situ. Discrepancies with regards to in situ conditions can therefore also be expected for the physical and mechanical pancake ice properties as obtained at $-10\,°\text{C}$ in the Cold Laboratory aboard the ship. Albeit, the latter does not affect the general depth trends of ice porosity and mechanical properties, it highlights the need to develop new effective methods facilitating testing in situ and minimize methodological errors.

The dynamic Young's and shear moduli measured for consolidated pack ice were substantially lower than reported in literature for first-year ice. The elastic properties of pancake ice were even lower in magnitude due to the high brine volume content and higher temperatures. A similar trend could be observed for the uniaxial compression strength which was found to be lower than reported by the only available source in literature for winter Antarctic sea ice, albeit land-fast (Urabe and Inoue, 1988a). For pancake ice, the uniaxial compression strength linearly increased with depth as opposed to the consolidated pack ice which decreased with depth as typically reported in literature (Han et al., 2015; Kermani et al., 2007).

Besides brine loss, the predominantly granular ice textures of the sampled Antarctic pancake and consolidated pack ice are most likely responsible that the obtained elastic and uniaxial strength properties are distinctly lower than reported in literature for columnar Arctic winter sea ice. In particular, the failure modes for columnar and granular ice are different which is more of a buckling nature for the former and shear failure for the latter as linked to the largest defect in statistical fracture mechanics. Lastly, the depth variation of mechanical properties is usually not accounted for in literature.

A distinct directional dependency of the elastic moduli in vertical and horizontal directions, respectively, has been found, in particular for consolidated pack ice. Interestingly, the elastic anisotropy of sea ice has not found much attention in literature and has been considered of minor significance. Clearly, besides temperature, internal ice composition and age have an at least equal influence on sea ice stiffness and strength. In particular, the remarkable vertical strength and stiffness profiles of pancake ice floes cannot be exclusively explained by temperature data but need to include salinity, brine volume and textural information as well. As such, an ambiguity arises if mechanical properties are solely related to ice porosity.

In summary, there is a complete lack of data concerning the mechanical behaviour of winter young and first-year sea ice in the Antarctic. The physical and mechanical sea ice properties obtained for young mostly granular Antarctic first-year ice exhibited distinct differences to measurements done on Arctic predominantly columnar sea ice in terms of compression strength, stiffness and directional dependency of elastic moduli warranting further studies to elucidate the influence of environmental conditions characteristic for the Antarctic marginal ice zone on the interrelation of textural and physical with the mechanical sea ice properties. Furthermore, the obtained data indicate a high variability which needs to be confirmed region and season-specific. Small-scale sea ice dynamics models can further help to advance understanding of the seasonal cycle of sea ice growth and retreat as linked mechanical phenomena such as inelastic collision and fracture of ice floes but rely on the accurate parameterisation of the mechanical sea ice parameters in particular.

*Data availability.* All experimental data part of this manuscript have been made available via a publically accessible data repository https://doi.org/10.25375/uct.14900361.

## Appendix A:  Pancake and consolidated pack ice core lists

## A1    Consolidated pack ice data

**Table A1.** Overview of pack ice core IDs, corresponding core names, testing designation, and date and time when cored.

| Core ID | Core Name | Test Designation | Date Cored | Time Cored |
| --- | --- | --- | --- | --- |
| 1 | M03-TM-01-A | Trace metal | July 27, 2019 | 10am-4pm (UTC) |
| 2 | M03-TM-02-A | Trace metal | July 27, 2019 | 10am-4pm (UTC) |
| 3 | M03-TM-03-A | Trace metal | July 27, 2019 | 10am-4pm (UTC) |
| 4 | M03-TM-04-A | Trace metal | July 27, 2019 | 10am-4pm (UTC) |
| 5 | M03-TM-05-A | Trace metal | July 27, 2019 | 10am-4pm (UTC) |
| 6 | M03-BGC-01-A | Isotope | July 27, 2019 | 10am-4pm (UTC) |
| 7 | M03-BGC-02-A | Isotope | July 27, 2019 | 10am-4pm (UTC) |
| 8 | M03-PHY-01-A | Temperature | July 27, 2019 | 10am-4pm (UTC) |
| 9 | M03-PHY-02-A | Temperature | July 27, 2019 | 10am-4pm (UTC) |
| 10 | M03-PHY-03-A | Temperature | July 27, 2019 | 10am-4pm (UTC) |
| 11 | M03-CPUT-01-A | Biology | July 27, 2019 | 10am-4pm (UTC) |
| 12 | M03-CPUT-02-A | Biology | July 27, 2019 | 10am-4pm (UTC) |
| 13 | M03-BIO-01-A | Bio-cultivation | July 27, 2019 | 10am-4pm (UTC) |
| 14 | M03-CT-01-A | MicroCT | July 27, 2019 | 10am-4pm (UTC) |
| 15 | M03-CT-02-A | MicroCT | July 27, 2019 | 10am-4pm (UTC) |
| 16 | M03-CP-01-A | Texture & fabrics | July 27, 2019 | 10am-4pm (UTC) |
| 17 | M03-CP-02-A | Texture & fabrics | July 27, 2019 | 10am-4pm (UTC) |
| 18 | M03-US-01-A | Elasticity | July 27, 2019 | 10am-4pm (UTC) |
| 19 | M03-US-02-A | Elasticity | July 27, 2019 | 10am-4pm (UTC) |
| 20 | M03-US-03-A | Elasticity | July 27, 2019 | 10am-4pm (UTC) |
| 21 | M03-DE-01-A | Compression strength | July 27, 2019 | 10am-4pm (UTC) |
| 22 | M03-DE-02-A | Compression strength | July 27, 2019 | 10am-4pm (UTC) |
| 23 | M03-DE-03-A | Compression strength | July 27, 2019 | 10am-4pm (UTC) |

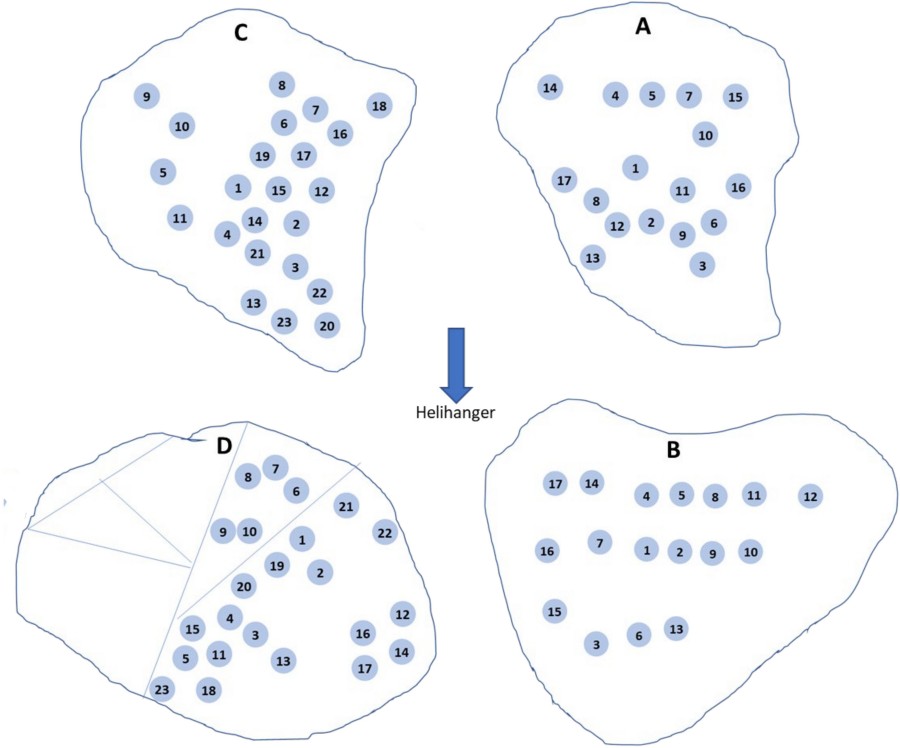

**Figure A1.** Coring layout for pancake ice floes A, B, C and D, respectively.

**Table A2.** Overview of core IDs of pancake A, corresponding core names, testing designation, and date and time when cored.

| Core ID | Core Name | Test Designation | Date Cored | Time Cored |
|---|---|---|---|---|
| 1 | M01-PHY-01-A | Temperature | July 28, 2019 | 10:50am (UTC) |
| 2 | M01-BGC-01-A | Isotope | July 28, 2019 | 11:05am (UTC) |
| 3 | M01-PHY-02-A | Temperature | July 28, 2019 | 11:10am (UTC) |
| 4 | M01-BGC-02-A | Isotope | July 28, 2019 | 11:12am (UTC) |
| 5 | M01-PHY-03-A | Temperature | July 28, 2019 | 11:15am (UTC) |
| 6 | M01-CPUT-01-A | Biology | July 28, 2019 | 11:30am (UTC) |
| 7 | M01-CPUT-02-A | Biology | July 28, 2019 | 11:32am (UTC) |
| 8 | M01-CT-01-A | MicroCT | July 28, 2019 | 11:50am (UTC) |
| 9 | M01-CT-02-A | MicroCT | July 28, 2019 | 11:52am (UTC) |
| 10 | M01-CP-01-A | Texture & fabrics | July 28, 2019 | 12:01pm (UTC) |
| 11 | M01-CP-02-A | Texture & fabrics | July 28, 2019 | 12:05pm (UTC) |
| 12 | M01-US-01-A | Elasticity | July 28, 2019 | 12:06pm (UTC) |
| 13 | M01-US-02-A | Elasticity | July 28, 2019 | 12:11pm (UTC) |
| 14 | M01-US-03-A | Elasticity | July 28, 2019 | 12:13pm (UTC) |
| 15 | M01-DE-01-A | Compression strength | July 28, 2019 | 12:18pm (UTC) |
| 16 | M01-DE-02-A | Compression strength | July 28, 2019 | 12:19pm (UTC) |
| 17 | M01-DE-03-A | Compression strength | July 28, 2019 | 12:22pm (UTC) |

**Table A3.** Overview of core IDs of pancake B, corresponding core names, testing designation, and date and time when cored.

| Core ID | Core Name | Core Type | Date Cored | Time Cored |
|---------|-----------|-----------|------------|------------|
| 1 | M01-PHY-01-B | Temperature | July 28, 2019 | 11:05am (UTC) |
| 2 | M01-BGC-01-A | Isotope | July 28, 2019 | 11:18am (UTC) |
| 3 | M01-PHY-02-B | Temperature | July 28, 2019 | 11:20am (UTC) |
| 4 | M01-BGC-02-B | Isotope | July 28, 2019 | 11:24am (UTC) |
| 5 | M01-PHY-03-B | Temperature | July 28, 2019 | 11:26am (UTC) |
| 6 | M01-CPUT-01-B | Biology | July 28, 2019 | 11:35am (UTC) |
| 7 | M01-CPUT-02-B | Biology | July 28, 2019 | 11:40am (UTC) |
| 8 | M01-CT-01-B | MicroCT | July 28, 2019 | 11:57am (UTC) |
| 9 | M01-CT-02-B | MicroCT | July 28, 2019 | 11:59am (UTC) |
| 10 | M01-CP-01-B | Texture & fabrics | July 28, 2019 | 12:35pm (UTC) |
| 11 | M01-CP-02-B | Texture & fabrics | July 28, 2019 | 12:47pm (UTC) |
| 12 | M01-US-01-B | Elasticity | July 28, 2019 | 12:50pm (UTC) |
| 13 | M01-US-02-B | Elasticity | July 28, 2019 | 12:52pm (UTC) |
| 14 | M01-US-03-B | Elasticity | July 28, 2019 | 1:38pm (UTC) |
| 15 | M01-DE-01-B | Compression strength | July 28, 2019 | 2:06pm (UTC) |
| 16 | M01-DE-02-B | Compression strength | July 28, 2019 | 2:09pm (UTC) |
| 17 | M01-DE-03-B | Compression strength | July 28, 2019 | 2:11pm (UTC) |

**Table A4.** Overview of core IDs of pancake C, corresponding core names, testing designation, and date and time when cored.

| Core ID | Core Name | Core Type | Date Cored | Time Cored |
|---------|-----------|-----------|------------|------------|
| 1 | M01-PHY-01-C | Temperature | July 28, 2019 | 1:05pm (UTC) |
| 2 | M01-BGC-01-C | Isotope | July 28, 2019 | 1:06pm (UTC) |
| 3 | M01-PHY-02-C | Temperature | July 28, 2019 | 1:18pm (UTC) |
| 4 | M01-BGC-02-C | Isotope | July 28, 2019 | 1:22pm (UTC) |
| 5 | M01-PHY-03-C | Temperature | July 28, 2019 | 1:26pm (UTC) |
| 6 | M01-TM-01-C | Trace metal | July 28, 2019 | 1:47pm (UTC) |
| 7 | M01-TM-02-C | Trace metal | July 28, 2019 | 1:53pm (UTC) |
| 8 | M01-TM-03-C | Trace metal | July 28, 2019 | 1:56pm (UTC) |
| 9 | M01-TM-04-C | Trace metal | July 28, 2019 | 2:14pm (UTC) |
| 10 | M01-TM-05-C | Trace metal | July 28, 2019 | 2:18pm (UTC) |
| 11 | M01-CP-01-C | Trace metal | July 28, 2019 | 2:23pm (UTC) |
| 12 | M01-CPUT-01-C | Biology | July 28, 2019 | 3:40pm (UTC) |
| 13 | M01-CPUT-02-C | Biology | July 28, 2019 | 3:42pm (UTC) |
| 14 | M01-CT-01-C | MicroCT | July 28, 2019 | 3:44pm (UTC) |
| 15 | M01-CT-02-C | MicroCT | July 28, 2019 | 3:45pm (UTC) |
| 16 | M01-CP-02-C | Texture & fabrics | July 28, 2019 | 3:47pm (UTC) |
| 17 | M01-US-01-C | Elasticity | July 28, 2019 | 3:50pm (UTC) |
| 18 | M01-US-02-C | Elasticity | July 28, 2019 | 3:51pm (UTC) |
| 19 | M01-US-03-C | Elasticity | July 28, 2019 | 3:53pm (UTC) |
| 20 | M01-DE-01-C | Compression strength | July 28, 2019 | 3:57pm (UTC) |
| 21 | M01-DE-02-C | Compression strength | July 28, 2019 | 3:58pm (UTC) |
| 22 | M01-DE-03-C | Compression strength | July 28, 2019 | 4pm (UTC) |

**Table A5.** Overview of core IDs of pancake D, corresponding core names, testing designation, and date and time when cored.

| Core ID | Core Name | Core Type | Date Cored | Time Cored |
|---|---|---|---|---|
| 1 | M01-PHY-01-D | Temperature | July 28, 2019 | 1:13pm (UTC) |
| 2 | M01-BGC-01-D | Isotope | July 28, 2019 | 1:16pm (UTC) |
| 3 | M02-BGC-01-D | Isotope | July 28, 2019 | 1:30pm (UTC) |
| 4 | M01-PHY-02-D | Temperature | July 28, 2019 | 1:35pm (UTC) |
| 5 | M01-PHY-03-D | Temperature | July 28, 2019 | 1:40pm (UTC) |
| 6 | M01-TM-01-D | Trace metal | July 28, 2019 | 2:34pm (UTC) |
| 7 | M01-TM-02-D | Trace metal | July 28, 2019 | 2:36pm (UTC) |
| 8 | M01-TM-03-D | Trace metal | July 28, 2019 | 2:39pm (UTC) |
| 9 | M01-TM-04-D | Trace metal | July 28, 2019 | 2:42pm (UTC) |
| 10 | M01-TM-05-D | Trace metal | July 28, 2019 | 2:44pm (UTC) |
| 11 | M01-CPUT-01-D | Biology | July 28, 2019 | 2:52pm (UTC) |
| 12 | M01-CPUT-02-D | Biology | July 28, 2019 | 2:55pm (UTC) |
| 13 | M01-CT-01-D | MicroCT | July 28, 2019 | 2:59pm (UTC) |
| 14 | M01-CT-02-D | MicroCT | July 28, 2019 | 3:02pm (UTC) |
| 15 | M01-US-01-D | Elasticity | July 28, 2019 | 3:07pm (UTC) |
| 16 | M01-US-02-D | Elasticity | July 28, 2019 | 3:11pm (UTC) |
| 17 | M01-US-03-D | Elasticity | July 28, 2019 | 3:14pm (UTC) |
| 18 | M01-CP-01-D | Texture & fabrics | July 28, 2019 | 3:19pm (UTC) |
| 19 | M01-CP-02-D | Texture & fabrics | July 28, 2019 | 3:20pm (UTC) |
| 20 | M01-DE-01-D | Compression strength | July 28, 2019 | 3:22pm (UTC) |
| 21 | M01-DE-02-D | Compression strength | July 28, 2019 | 3:26pm (UTC) |
| 22 | M01-DE-03-D | Compression strength | July 28, 2019 | 3:30pm (UTC) |
| 23 | M01-CPUT-03-D | Biology | July 28, 2019 | 3:35pm (UTC) |

# Appendix B: Core segmentation for elasticity and uniaxial compression strength testing

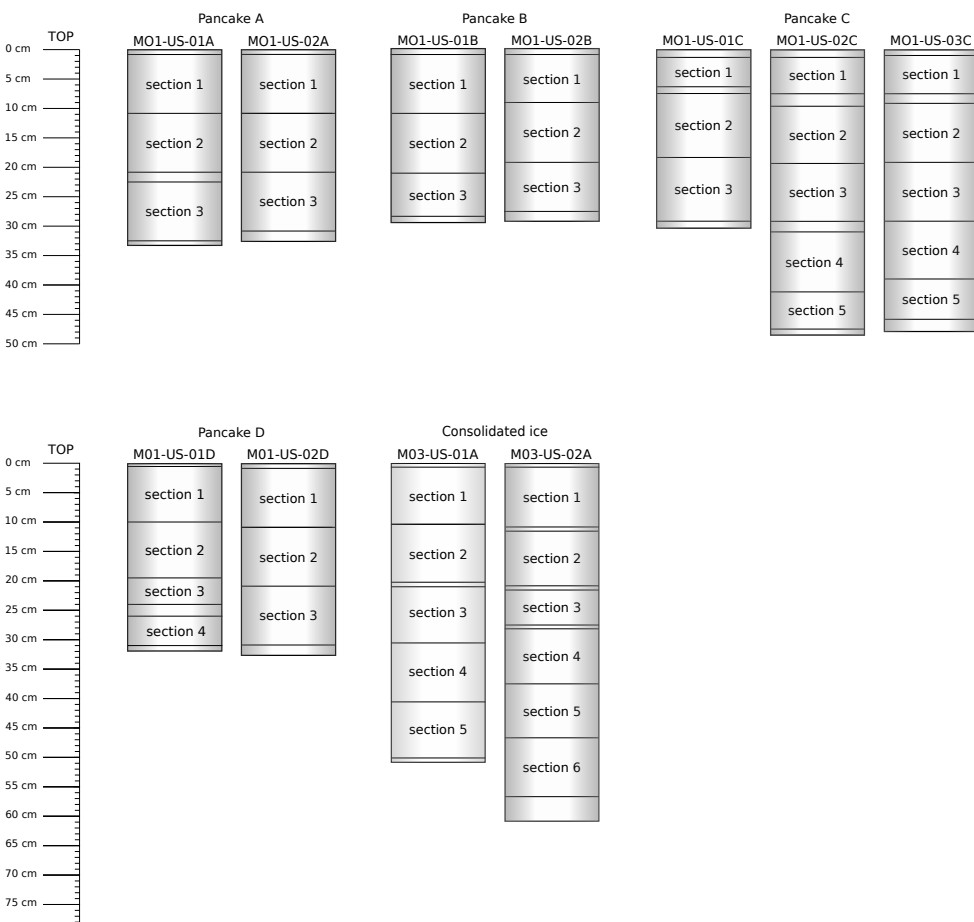

**Figure B1.** Schematic of the segmentation of seven pancake ice and two pack ice cores subjected to ultrasound testing with section dimensions and corresponding section numbering.

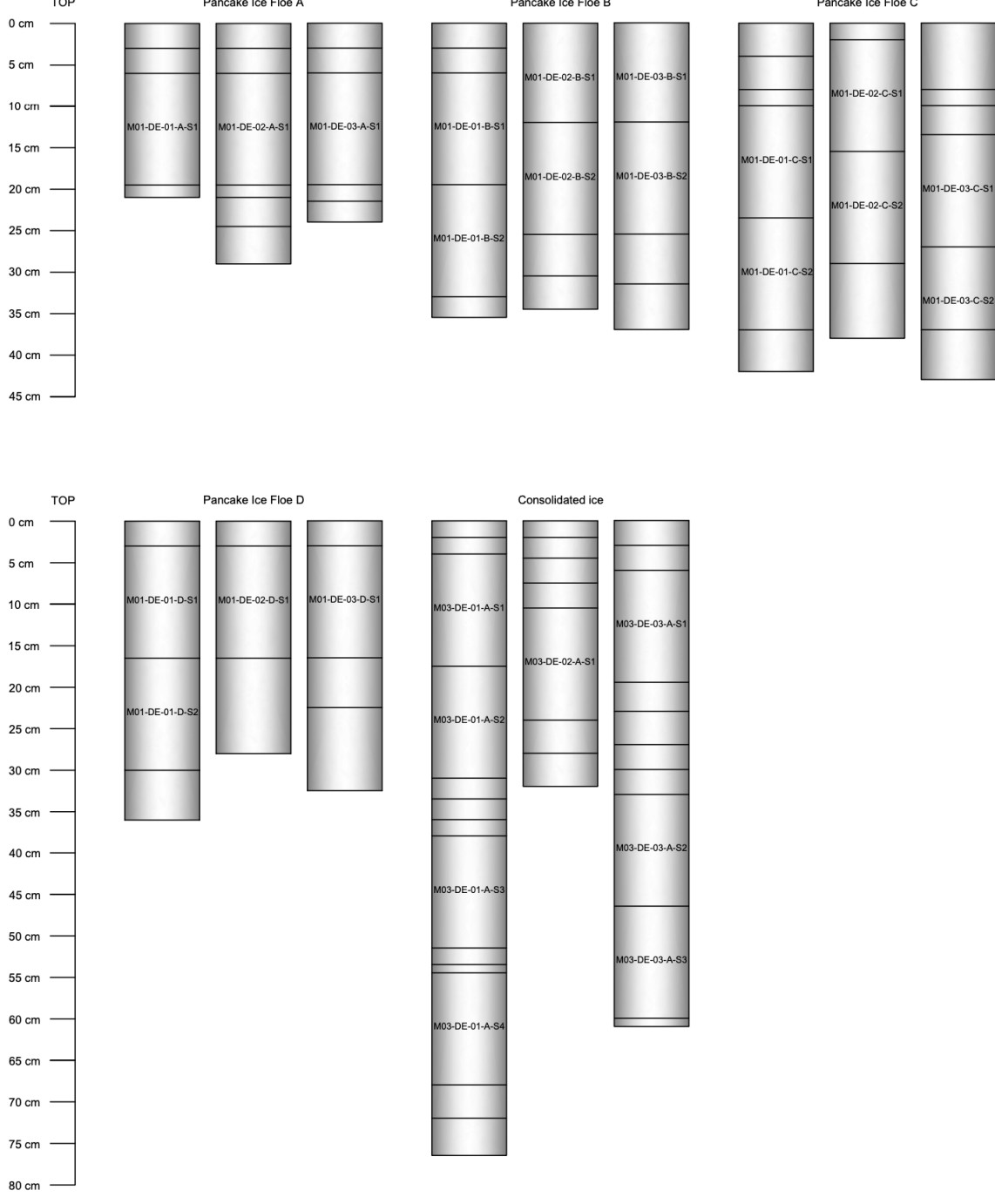

**Figure B2.** Schematic of the segmentation of twelve pancake ice and three pack ice cores subjected to compression testing with sample dimensions and corresponding core names.

*Author contributions.* Conceptualization by SS, DL, KM, TM, and MV. Morphological analysis methodology developed by SJ and TR, ultrasound testing methodology developed by SS, KM and RM, uniaxial compression testing methodology developed by DL, TM and FP. Investigation and data curation by SS, RA, AC, EH, SJ, KM RM, TM, EM, FP, TR and CS. Formal analysis by SS, RA, AC, SJ, DL, KM, RM, TM and FP. Funding acquisition by SS, DL, JS and MV. The original draft was written by SS with assistance from all co-authors.

*Competing interests.* The authors declare that the research was conducted in the absence of any commercial or financial relationships that could be construed as a potential conflict of interest.

*Acknowledgements.* This research has been funded by the National Research Foundation of South Africa (NRF) (Grant Nos. 104839, 105858 and 118745). Opinions expressed and conclusions arrived at are those of the author and are not necessarily to be attributed to the NRF. SS, RA, AC, EH, RM, EM, TR and MV acknowledge support from the South African National Antarctic Programme (SANAP).

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
