# Peer review of "Physical and mechanical properties of winter first-year ice in the Antarctic marginal ice zone along the Good Hope Line"

_The Cryosphere, 2021_

## Author Response (AR1)

**Physical and mechanical properties of winter first-year ice in the Antarctic marginal ice zone along the Good Hope Line; manuscript ID: tc-2021-209**

Report of Referee # 1 and Responses by Authors

*The authors would like to thank the reviewer for the meticulous reading of the manuscript, the comments and suggestions, which helped to improve the paper. All recommendations are addressed within the manuscript as highlighted in yellow and commented in the following using red font.*
* * *
The manuscript provides a good analyses on the physical and mechanical properties of winter first-year ice. And four pancake ice floes were collected using ship's crane, different from pervious studies using ice cores. So the results are interesting to me, and I have some comments as below.

1. L33: what is the full name of SO?

   **Response:**
   *SO is an abbreviation for Southern Ocean which has been introduced in line 16.*

2. L34: in my opinion, sea ice morphology is the exterior properties such as ridge, size, shape of sea ice, while the grain size, crystallographic texture and fabrics are interior properties.

   **Response:**
   *Agreed. Albeit the terminology "morphology" is used for internal ice texture and fabrics in literature, e.g. Petrich, C. and Eicken, H., 2017, Overview of sea ice growth and properties. Sea ice, pp.1-41, it is ambiguous and has been replaced throughout the revised manuscript as suggested.*

3. L98: -10 degrees Celsius is a bit higher, though the cores were placed in a horizontal position, brine drainage may be strength than at a cold temperature.

   **Response:**
   *Agreed, brine drainage is a factor influencing the results, in particular the initial drainage after the pancake is lifted out of the water. -10 degrees Celsius lab temperature has been considered as a good compromise by the authors when operating on a ship in the marginal ice zone. The atmospheric temperature is highly variable, even during a single day, and the sea ice temperature itself is highly variable spatially as well. Additional elaborations have been added on the reasons and limitations of this choice, see lines 114 seqq., 118 seqq. and 254 seqq. in revised manuscript.*

4. L122: more detailed information on the ice salinity measurements are needed. Were cores cut immediately after they were extracted or how long between the salinity measurements and core drilling? It affects the estimation of ice salinity measurement accuracy.

   **Response:**
   *Due to logistical constraints operating in the field, only the temperature measurements were immediately done after core extraction, as it is the most time-critical property. The cores were subsequently put in plastic sleeves, sealed and horizontally stored for up to 2 hours at*

*environmental temperatures when in the field or at -10 degrees Celsius aboard the ship in the cold lab until processed, depending whether they were extracted overboard or on deck. The latter refers to pancake floe ice which was immediately taken to the cold lab and processed within 30 minutes after the temperature measurements were done. Additional details have been added, see lines 114 seqq., 118 seqq. and 145 seqq. in the revised manuscript.*

5. Necessary error analysis on the ice salinity, density, modulus and compressive strength are absent.

   **Response:**
   *An estimate of error in the measurement and registration of temperature, salinity, density, elastic properties and compression strength is each less than 9%, which is significantly less than errors introduced by the variation of the ice structure and properties. A clarifying statement in this regard has been added and more elaborations on error sources and uncertainties for all presented data have added as well to the revised manuscript, see lines lines 113 seq., 145 seqq., 191 seqq., 199 seqq., 210 seqq., 254 seqq., 260 seqq., 263 seq., 342 seqq. and 369 seqq.*

6. Give a brief introduction on the stiffness of the compression machine.

   **Response:**
   *The PLT-2W has a virtually infinite stiffness, because it automatically corrects for the frame deformation based on the location of the top cross-head and as a function of the axial load. An explanatory sentence has been added, see lines 199 seqq. in revised manuscript.*

7. L219: what is the probable reason for the high density at the top layer.

   **Response:**
   *The reason is most likely due to the small sampling size and measuring errors. Determining sea ice density is often reported as a difficult task, especially on sea. The precise measuring of dimensions, the cutting and weighing was impaired when done on the ship at the edge of the marginal ice zone due to the significant swell. More elaborations have been added to remark on these issues. See lines 260 seq. and 263 seqq. in revised manuscript.*

8. Figure 4: Give the method how to determine the crystal size. Was plate spacing measured? How was the confidence interval determined?

   **Response:**
   *An average of 20 random crystals were selected per texture for measurement in a sample photograph and the average size and standard deviation was calculated using the scale on the photographs. Furthermore, a 95 percentile confidence interval test was performed over the data collected for each texture for statistical comparison. The plate spacing was not measured, because the sea ice samples were almost exclusively granular. More details have been added, see lines 167 seqq. in the revised manuscript.*

9. L256: a grammatical mistake, "there no" should be "there is no".

   **Response:**
   *Has been corrected in the revised manuscript.*

10. L271: are there references? It seems that the strength of granular ice is more than that of columnar ice.

**Response:**

*Both, the sampled consolidated pack ice and pancake ice, have almost exclusively granular textures. Young's modulus and strength magnitudes are generally higher in consolidated pack ice, see lines 315 seq. and 348 seqq., respectively, in the revised manuscript.*

*There are no references with regards to mechanical properties of winter Antarctic sea ice which can be used for comparison. In this sense, our data are absolutely unique which has been now highlighted, see lines 52 seqq. in revised manuscript.*

11. Figure 13: also give the quantitative dependency of modulus on brine volume. Why the relationship between young's modulus and brine volume for pancake ice is significant but not for pack ice in longitudinal direction.

    **Response:**

    *Relating the elastic properties measured at -10 degrees Celsius to recomputed relative brine volume data using the actual uniform -10 degrees Celsius ice temperature, a significant negative relation is now found for longitudinal and transverse direction for both ice types, see lines 333 seqq. in the revised manuscript.*

12. L317: it is questionable to assume negligible air volume fraction. As ice density are measured, therefore sea ice porosity can be determined by ice temperature, salinity, and density according to following references. And then compare it with Kovacs (1997).

    **Response:**

    *Agreed, a negligible air volume fraction cannot be assumed. This statement has been removed from the revised manuscript.*

    *As the accuracy of the density measurements was probably not optimal, the authors chose to use the empirical relation by Frankenstein and Garner (1967) which does not require knowledge of the sea ice density. These issues and our choice have been further elaborated on, see lines 254 seqq. and 260 seqq. in the revised manuscript.*

13. L338: Two "and" in the sentence.

    **Response:**

    *Has been corrected in the revised manuscript.*

14. Authors should emphasize the aim of this investigation on sea ice mechanical properties. Is it for engineering or geophysics. For engineering, effective modulus is more widely-used than elastic modulus; while for geophysics, it is necessary to state the application of elastic and shear modulus in the nature process or sea ice models.

    **Response:**

    *Our group is studying sea ice dynamics by means of satellite image analysis, ice-tethered instruments and numerical modelling. As pointed out by the review comment above, the dynamic elastic moduli are required for the latter. A sentence clarifying this has been added, see lines 172 seq. in the revised manuscript.*

**Physical and mechanical properties of winter first-year ice in the Antarctic marginal ice zone along the Good Hope Line; manuscript ID: tc-2021-209**

Report of Referee # 2 (editor) and Responses by Authors

*The authors would like to thank the reviewer for the meticulous reading of the manuscript, the comments and suggestions, which helped to improve the paper. All recommendations are addressed within the manuscript as highlighted in yellow and commented in the following using red font.*
* * *
This study reports physical and mechanical properties of sea ice from a very few floes in the marginal ice zone. While the dataset is small, and the number of floes selected possibly too small to know if they are representative, there are two aspects that make these measurements notable. First, there are almost no prior observations of mechanical properties for Antarctic sea ice, and I believe these are the first for the MIZ. That alone makes these valuable. Second, while such few cores for the physical properties do not add much to the fairly extensive prior observations from the literature, they are unique in that so many cores were taken from a few pancake floes. For these reasons, there is enough here to be worth publishing.

There are a few important areas where the manuscript can be improved. I have two main suggestions; one on the presentation, and the second is technical.

1. The measurements are not placed in sufficient context and the **importance of the results is not discussed in much detail**. The **introduction is somewhat short** and does not provide enough background on prior work. It would also be helpful to provide **more background on prior similar mechanical measurements up front** to help highlight the lack of such measurements in the MIZ, and to provide more background explaining **how your measurements can inform sea ice mechanical models**. This may be less clear for large-scale ice rheology, but I think your measurements are more directly relevant for things like rafting and ridging, and particularly for pancake-pancake interactions, wave propagation, and fracture of pancakes. You could also **summarize/reference prior observations of sea ice properties made in the MIZ**. In addition to the ones you mention, numerous cores in the MIZ were taken on various **Jeffries cruises between 1994-2000**, and most recently **Ackley et al., 2020**. Finally, more discussion of the potential importance and impact of your results in the conclusion would help.

   **Response:**
   *An additional paragraph has been added to the revised manuscript elaborating on the related measurements and observations reported in literature. Furthermore, the scarcity of mechanical properties with regards to Antarctic sea ice and their complete lack with regards to elastic properties of Antarctic winter first-year sea ice, see lines 48 seqq. in the revised manuscript.*

   *As suggested, the need and relevance of mechanical ice properties, in particular, to inform detailed small scale sea-ice dynamics models has been highlighted, see lines 54 seqq. in the revised manuscript.*

   *Comparisons with relevant in observations in literature have been provided, see lines 232 seq., 249 seqq. and 290 seq. in the revised manuscript.*

2. The interpretation of the **results with respect to the brine volume**, which will be significantly **different between in situ and when measured in the cold lab**, is suspect, and not sufficiently discussed. This point is critical, as I believe some of the results (figure 13) are then incorrect (see specific comments below). The potential impact of the results could also be better described.

**Response:**

*Direct brine volume measurements are difficult in situ at best and the indirect determination based on existing empirical relations (Frankenstein and Garner 1967 or Cox and Weeks 1983) is generally hampered by brine drainage after sea ice extraction/coring as well as inaccuracies of onboard density measurements. For this reason, the relation by Frankenstein and Garner 1967 was used which does not depend on ice density but is solely based on ice bulk salinity and temperature. Whereas brine drainage for consolidated pack ice mostly effects its bottom layer, it can be expected that the generally high porosity of pancake ice floes leads to more significant drainage effects. Additional comments have been added with regards to brine loss and its impact when relating relative brine volume to mechanical properties, see lines 118 seqq., 147 seqq. and 254 seqq. in the revised manuscript.*

*Furthermore, as ice stiffness and strength have been measured in the lab at -10 degrees, both results are now plotted in Figures 14 and and 16 with respect to brine volume at -10 degrees Celsius ice temperature. Figure 11 has been added to show the relative brine volume vs. depth for a uniform ice temperature of -10 degrees Celsius. The statistical analysis has been updated accordingly, see lines 297 seqq., 333 seqq. and 342 seqq. in the revised manuscript.*

3. Unfortunately, the **differences among cores for different locations** within the individual pancakes is not explored. I would urge the authors to explore this, both for the salinity and mechanical properties, if possible, as there have been almost no descriptions of such variations before (I believe one or two papers by Wadhams has some information on spatial variations in salinity).

**Response:**

*The authors agree that a detailed analysis of distribution of properties within a single pancake ice floe would be very important to investigate. Unfortunately, there was a large interest from different groups part of the expedition to study a wide spectrum of pancake ice floe properties, e.g. temperature, salinity, trace metal, biomatter inclusions, elasticity, strength, crystallography etc. Accordingly, only 3 cores per floe could be allocated for each, salinity, elastic and compression strength properties. As such, it was not possible to confidently report on the actual distribution of each property within a single pancake ice floe.*

Specific comments:

a) Line 17: I normally think of "sea ice properties" as intrinsic, and so not something climate models would predict (e.g., models don't predict ice strength). I think you mean things like extent here.

**Response:**

*Correct, we are referring to predicted sea ice properties in terms of extent, concentration and thickness as affected by drift and thermodynamics. The sentence has been rephrased accordingly, see line 16 in the revised manuscript.*

b) Line 20-22: sea ice mechanics is not my core area of expertise, but I think this is too strong a statement. While there is considerable debate over issues of scale in sea ice mechanical properties, the large-scale emergent properties do have some physical meaning. i.e. P* is still related to compressive strength, and I believe Schulson (JGR, 2004) has argued that fracture patterns are similar across scales and suggests that mechanical properties may be scale independent. Rheologies do have empirical parameters, but many do have physical meaning (some, however, are numerical conveniences). You might also add a couple refs here, as there are a variety of sea ice rheologies.

**Response:**
*The scale dependency of sea ice deformation has been a field of active research until now. Differences between Arctic and Antarctic sea ice dynamics further add complexity to this. Squire 2018 summarized that meso and large-scale sea ice dynamics models are phenomenological approaches and their parameters do not represent actual physical ice properties. They are only fitted average values of a large enough region containing varying sea ice types. Moreover, these averaged values also incorporate a whole range of mechanical phenomena in combination such as floe collision dynamics, form drag of interstitial grease ice on pancake ice floes, skin drag (see e.g. another paper by the authors on detailed small-scale modelling of pancake ice floe dynamics in the MIZ: Marquart et al. 2021).*

*The authors tend to agree with the conclusions by Squire based on their own observations during the SCALE Winter Cruises to the Antarctic MIZ (2017 and 2019) where they experienced a very heterogeneous and mobile sea ice composition comprising a mixture of solid pancake ice floes and fluid-like grease ice, varying ice concentration, leads with open water etc.*

*The scale-dependency of sea ice deformation has been also discussed by Rampal et al. 2008 and Dansereau et al. 2016 where it was concluded that it cannot be exclusively described by a viscous-plastic rheology, e.g. as sea ice drift on the scale of less than 10 km is only accurate to some degree and fails to reproduce sea ice deformation at finer scales.*

*Also, brittle sea ice mechanics and fracture is linked to a wide range of spatial and temporal scales and highly intermittent requiring coupling and tailored scaling laws to link theses scales (see "Linking scales in sea ice mechanics" (2017), Weiss and Dansereau). The issue of scale-dependency sea ice models has been further elaborated on see lines 20 seqq. in the revised manuscript.*

*Dansereau, V.; Weiss, J.; Saramito, P.; Lattes, P. A Maxwell elasto-brittle rheology for sea ice modelling. Cryosphere 2016, 10, 1339-1359.*
*Marquart, R.; Bogaers, A.; Skatulla, S.; Alberello, A.; Toffoli, A.; Schwarz, C. and Vichi, M., 2021. A computational fluid dynamics model for the small-scale dynamics of wave, ice floe and interstitial grease ice interaction. Fluids, 6(5), 176.*
*Rampal, P.; Weiss, J.; Marsan, D.; Lindsay, R.; Stern, H. Scaling properties of sea ice deformation from buoy dispersion analysis. J. Geophys. Res. Ocean. 2008, 113.*
*Squire, V. A. A fresh look at how ocean waves and sea ice interact, Philosophical Transactions of the Royal Society A: Mathematical, Physical and Engineering Sciences 376 (2129) (2018) 20170342.*
*Weiss, J.; Dansereau, V. Linking scales in sea ice mechanics, Philosophical Transactions of the Royal Society A: Mathematical, Physical and Engineering Sciences 375 (2086) (2017) 20150352.*

c) Figure 1: This is a bit hard to read because of the color choices. At least highlight the two sites where cores were taken and reported in this paper with a different color. You might consider alongside this figure a location map that shows where this was in relation to the rest of the Weddell Sea region.

**Response:**

*Figure 1 has been improved including a large-scale location map in the revised manuscript.*

d) Line 71-75: Can you define what is meant by young ice here? Is it the WMO definition?

**Response:**

*With reference to WMO (code 3739), ice age ID 3 applies for the station where the pancake ice ice was collected (predominantly new and/or young ice with some first-year ice) and ice age ID 5 applies for the station where consolidated pack ice was collected (all thin first-year ice (30 - 70 cm thick). This clarification has been added to the manuscript, see lines 95 seq. in the revised manuscript.*

e) Lines 85-90: only a point on style here, but listing the tools is a bit of an odd way to present this. Normally, you'd mention the specific tools when their use is mentioned in the following paragraph.

**Response:**

*A: Has been changed in the revised manuscript as suggested.*

f) Line 121-124: Measuring salinity for young, warm ice is challenging, because the brine drains so quickly. Normally, one would measure salinity immediately upon sampling. It is not clear how much time passed between taking the core and cutting on the band saw. In any case, you should note that some salt may have drained, so your salinities may in some case be low. This could impact density measurements as well.

**Response:**

*The processing time of core samples used to determine the bulk salinity of ice was less than an hour. However, consolidated pack ice cores stayed in the field for up to 2 hours before being processed. More clarification has been added, see lines 113 seqq., 118 seqq., 145 seqq. and 254 seqq. in the revised manuscript.*

g) Line 125: note it is the absolute value of the temperature that is used in (1).

**Response:**

*Has been noted, see line 152 in the revised manuscript.*

h) Line 137: what is a thermal macrotome? Thin sections are normally done on a sliding microtome, so some description of the instrument would be useful here if it is different.

**Response:**

*The authors used a thermal macrotome to produce thin sections for cross-polarisation viewing which they constructed themselves. This device uses the concept of heat being passed through a nickel-chrome wire that slices ice into thin sections. It is able to cut 10 cm-long and 9 cm-wide core segments into slices of 1 mm or less thick. More details to this custom-made macrotome have been provided, see lines 163 seqq. in the revised manuscript.*

i) Section 3.1: It would help to compare these salinity profiles to prior observations from the MIZ. For example Eicken, 1992 is the most extensive study on salinity profiles in Antarctic sea

ice, and defined several canonical shapes; how do yours compare? Also, see Tison et al. 2020, who report salinities and brine volumes for young ice in the Ross Sea, with several very high salinities reported.

**Response:**

*A comparison with existing literature has been added to the manuscript to provide more context, see lines 232 seq.*

j) For bulk density, please provide more details on how the measurements were made. This is a difficult measurement to make accurately based on weighing cut pieces (and may be affected by brine drainage). You state you have some implausible values; doesn't this imply that your confidence intervals should be broader in your figures 8 and 9? i.e. the error is not just to scatter in the fit, but also due to potential inaccuracies in the measurement. They look too small for pancake ice. How do these densities compare to prior observations? Density is a quite useful property to know, and few have been reported in the Antarctic, so it would be quite useful to have a more careful estimate of your confidence in these numbers in case someone uses them.

**Response:**

*The density measurements are indeed highly problematic when done aboard a ship in the winter MIZ close to the open ocean. Also, the errors in the determination of specimen dimensions and volume from cored specimens due to flaws in terms of shape and surface composition as well as brine loss play a role and lead to unavoidable inaccuracies computing the density.*

*The graphical representation of the 90th percentile confidence intervals have been computed with the help of the Seaborn package (https: // seaborn. pydata. org/ ) and have been double-checked. The relatively low pancake ice density is plausible when considering brine loss during floe extraction from the ocean. For the consolidate pack ice density, another issue is the relatively small sample size which results in unrealistic values for the top part in particular.*

*More details have been provided , as to how the measurements were obtained using a Ross tape measure and a hand-held scale, see lines 244 seqq. in the revised manuscript. Further comments have been added with regards to the issues negatively affecting the density measurement accuracy and the relatively low pancake ice density values, see lines 260 seqq. and 263 seqq.*

k) Figures 6&7: it might be more useful to show temperature, salinity, and brine volume alongside each other in each figure, as this might help the interested reader in understanding the mechanical results. This is just a presentation style, so up to you.

**Response:**

*The authors thank the reviewer for this suggestion. Indeed different combinations were tried, because of the high number of measurements and the large ranges involved. The figures for the different cores and stations are quite small. Merging all 3 properties into 1 graph led to decreased legibility. The authors therefore would prefer to maintain the current layout.*

l) Line 226-229: this is misleading. First snow ice is often orbicular granular as well (which is why it is difficult to identify by morphology alone). It is superimposed ice that is clearly polygonal. Second, this is not the primary means of snow ice formation (though perhaps it is in the MIZ, where it is dynamic and snow is not deep). Snow ice is usually thought of as forming from seawater percolation up through the ice when under sufficient snow loading.

**Response:**

*It was indeed not possible from our analysis to differentiate between ice formed from frazil ice as opposed to meteoric ice and the specific ratio of both. Isotopic analyses have been delayed during the pandemic and they are planned for this year. We therefore only refer to it as granular ice textures. We agree with regards to the origin of snow ice formation and textural ambiguities of granular ice. The relevant sentences have been rephrased accordingly, see lines 271 seqq. in the revised manuscript.*

m) Figure 11 and 12: what is the purpose of plotting these versus depth? It doesn't look like there is any significant relationship, and is certainly more due to brine volumes, etc, which you have not properly captured (see next point).

**Response:**

*As correctly pointed out by the reviewer, there is significant uncertainty with regards to the porosity data obtained and therefore, a direct relation of porosity with elastic properties carries the same uncertainty. As such, the authors thought it meaningful to provide all physical and mechanical properties also independent of each other in form of their vertical spatial distribution. In this way, it was possible to subsequently cross-referenced and validated all ice properties against each other, as there is an interlink between the vertical textural, porosity and mechanical properties.*

*Furthermore, the directional dependency of mechanical properties can only be unique identified based on their spatial distributions.*

*Lastly, there is a need for the detailed spatial distribution of mechanical properties, such as elasticity and strength for small-scale modelling of the mechanical behaviour of ice floes (e.g. inelastic floe collision). In this sense, the spatial distribution of mechanical properties is not only required in horizontal directions (as requested before by the reviewer) but also in the vertical direction.*

n) Line 282-286, figure 13, 16, and elsewhere: It is not clear if you have plotted against the brine volume in situ (i.e. based on the initial core temperature and salinity) or in the lab (where the cores would have cooled somewhere close to the -10C storage temperature by the time these tests were performed. Based on the values plotted, It seems like it is the former for Figure 13, but the latter for Figure 16. This is important, because your cores were generally much warmer when sampled. If it is the former, then it is probably quite misleading. Ice with salinity of 7 ppt at -10C will have $V_b$ 4%. This puts all your values for Youngs modulus well below Langleben and Pounder's. But more importantly, it affects your interpretation of the actual properties of the ice in the ocean, because the brine volumes will be much higher there in most cases. This needs to be explained and discussed at some length.

**Response:**

*The temperature measurements were done in situ immediately after core extraction, whereas the bulk salinity measurements were obtained from melted core segments cut within 30 min (2.5 hrs) after extraction in the cold lab at -10 degrees Celsius. As for the bulk salinity, only drainage but not temperature affect its magnitude. As mentioned before, the authors only observed substantial initial drainage during ice collection. In this sense, the physical properties temperature, salinity and thus, relative brine volume (computed from temperature and salinity values) are relatively close to in situ conditions but have been affected by drainage at extraction. As such, the actual relative brine volume before ice extraction from the ocean can be assumed*

*to have higher magnitudes.*

*On the other hand, the mechanical properties were obtained in the cold lab at -10 degrees within 2 to 3 hours after extraction, i.e. the mechanical properties relate to sea ice at -10 degrees Celsius whereas relative brine volume displayed in figure 8 to the in situ temperature distribution. In this sense, the reviewer is absolutely correct that there is an ambiguity in the mechanical properties plotted vs. brine volume. Therefore, the relative brine volume distributions for all cores has been recomputed using the relation by Langleben and Pounder 1963 with uniform -10 degrees Celsius ice temperature so that they directly correspond to the ice conditions when the mechanical testing was done, see Fig. 11 in the revised manuscript. The graphs illustrating mechanical properties vs. brine volume have been updated, see Figs. 14 and 17 in the revised manuscript.*

o) Line 293: This statement is inconsistent with line 303, which states that Kivamaa and Kosloff did such measurements in the Weddell Sea.

**Response:**
*Only Urabe and Inoue (1988) obtained Antarctic sea ice strength measurement during **winter** (albeit land-fast). Kivimaa and Kosloff (1994) also obtained Antarctic sea ice strength measurements, but during spring. In this sense, only Urabe and Inoue as well as ourselves obtained winter Antarctic sea ice strength measurements. This difference and unique feature has been now emphasized in line 345 in the revised manuscript.*

p) Conclusion: can you elaborate on your results in terms of what they might imply for sea ice mechanical modelling, etc?

**Response:**
*Agreed. The obtained data on mechanical properties of pancake ice floes in the Antarctic winter MIZ will help to parameterize realistic small-scale sea ice dynamics models with respect to aspects concerning the influence of pancake ice floe deformation on the inelastic collision restitution and pancake floe fracture due to collision. Both are expected to influence sea ice formation, sea ice drift and wave attenuation.*

*The obtained measurements of anisotropic elastic material properties of consolidated pack ice, in particular, are unique, because they also comprise their directional dependency (anisotropy) in vertical and horizontal directions, respectively. This aspect has been considered insignificant in literature and is usually not reported on. Where applicable, the complete set of anisotropic elastic material properties, however, is required when realistically modelling sea ice deformation. Another paragraph has been added, see lines 412 in the revised manuscript.*

---

## Author Response (AR2)

**Physical and mechanical properties of winter first-year ice in the Antarctic marginal ice zone along the Good Hope Line; manuscript ID: tc-2021-209**

**Report of Editor and Responses by Authors**

*The authors would like to thank the reviewer for the meticulous reading of the manuscript, the comments and suggestions, which helped to improve the paper. All recommendations are addressed within the manuscript as highlighted in yellow and commented in the following using red font.*
* * *
The manuscript is much improved and addressed almost all the comments adequately. I think there are two main points that could use some minor improvements.

1. Your results are quite different now that you have used the brine volumes during testing. There is some discussion of why these results might be different in the text, but I would have appreciated a bit more discussion, particularly in the conclusion about why this might be (e.g. differences in structure, potential differences in in air volumes, or maybe even differences in the technique, in that prior work tended not to resolve depth variations, etc). One thing you might also try to address is whether the differences you see are real differences, or statistical. E.g. in Figure 14, you see differences between pancake and consolidated ice for Young's modulus, but at least for some brine volumes, I am not sure these are that statistically significant given the spread in your values (the 90% confidence you show is for the line fit, but I believe this does not represent the range that 90% of your values at a given brine volume fall within). Likewise, can you elaborate a little more on the differences in compressive strength with previous work? Is it measurement technique (lab vs insitu)? Is it texture or ice type? Etc. I think just a few sentences in the Results and Discussion and/or Conclusions would help readers better understand what they should take away from your results.

   **Response:**

   *We have expanded more on results and findings in the conclusion of the revised manuscript. As suggested, we provided more details on the linkages between textural, physical and mechanical properties as well as differences between Arctic and Antarctic sea ice observations.*

   *You are correct. By definition, the 90% confidence interval is not the range that contains 90% of the values but the range of values where one can be 90% certain to contain the true mean of the population.*

2. My one comment that was not really addressed was about what conclusions that you can draw about mechanical properties in situ from your lab measurements. Your in situ brine volumes are substantially higher than in the lab, but there is some overlap. i.e., a typical brine volume seems to be about 10% in situ (for a bulk brine volume for the whole core, this seems pretty good, which may be a reasonable number to use for the mechanical properties of a whole floe). In the lab, you have a few values for pancake ice in this range. Do you think then these would provide reasonable values for others to use for the MIZ? Unfortunately, you don't have similar values for consolidated ice, and your curves would extrapolate to

unrealistic values at 10% brine volume. I think some discussion of this is needed in the conclusion.

**Response:**

*We have further discussed in the conclusion of the revised manuscript the found differences of mechanical sea ice properties compared to literature due to differences regarding ice origin, type and texture as well as due to methodological challenges with the mix of in situ/ex situ testing. We also emphasized the importance to relate mechanical sea ice properties not only to its porosity but rather as linked to the combination of ice temperature, bulk salinity and textural characteristics. Lastly, we pointed out that the obtained data indicate a high variability which warrants further studies and confirmation specific to region and season.*

Minor comments:

1. Check the formatting of references cited within the text. There seem to be lots of errors generated by your referencing software, and the in-text citation format is not standard for the journal.

   **Response:**

   *All citation errors have been corrected in the revised manuscript.*

2. Figure 1 - this is much improved. The inset is clear enough, I suppose, but think it could be a bit better if the scale was such that you could see more of the coastline of the Antarctic so it was more instantly recognizable to the reader.

   **Response:**

   *The figure has been further improved as suggested in the revised manuscript.*

3. Figure 14 and text - note that the equation you attribute to Langelben and Pounder is actually from Langleben, 1962 for first year ice. In Langelben and Pounder there results fall below this line (though not as much as yours) and are for multiyear sea ice. Note that they measure E in situ, but the paper is not very clear on how they treat brine volume - presumably the in situ value, but it will vary from top to bottom, and their technique gives a bulk value of E, so could help explain why your values do not match theirs?

   **Response:**

   *As mentioned above, we believe that the textural differences between columnar Arctic sea ice and granular Antarctic sea ice are of significance being partly responsible for the differences regarding the Young's modulus besides the brine loss. Another clarifying sentence has been added, see lines 340 seqq. and also in the conclusion of the revised manuscript.*

4. Figure 17 - it would be nice to have a little more discussion of these results, since they appear to be quite different the Kovacs for low brine volume. Maybe it is all due to brine loss (which you could perhaps estimate), but you'd expect that to be less for the lower brine volumes. One possibility is Kovacs used bulk brine volume using a single temperature, and you are using more discrete data (I think). But it could also be different textures and air volumes (particularly for pancake ice).

   **Response:**

   *Besides loss of brine volume, we believe that the difference regarding the uniaxial compression strength appears to be due to textural differences between columnar Arctic sea ice and granular*

*Antarctic sea ice which has been already mentioned in the manuscript, see lines 370 seqq.,
but has been further stressed and discussed in the conclusion of the revised manuscript.*

5. I still don't understand what you are trying to show in figure B1. All I see is grey blocks
of various lengths, and only M01-US-01D shows different sections labelled, but this is just
text on the grey block, so I can't even judge the thickness of these sections. Maybe you'd
be better off here with a table that lists the length of the sections and total core lengths for
each core?

**Response:**
*There seems to be an issue with certain PDF readers, as the figure shows the core segmenta-
tion with appropriate labelling and dimensions. I will upload both figures separately as well
for your reference.*